# CROSS-MODAL FEATURE DISENTANGLEMENT WITH CONTRASTIVE TASK ALIGNMENT FOR MULTI-MODAL IMAGE FUSION

## ABSTRACT

Multi-modal image fusion suffers from feature entanglement, where modality-specific, content-specific, and task-specific information becomes conflated in unified representation spaces, leading to suboptimal fusion quality and limited generalization. This paper proposes Cross-Modal Feature Disentanglement with Contrastive Task Alignment (CMD-CTA), a framework that addresses this fundamental challenge through mathematically motivated feature separation and semantic alignment, supported by both theoretical analysis under idealized assumptions and empirical evidence on real-world fusion benchmarks. The approach introduces two key innovations: (1) differentiable orthogonal feature decomposition that encourages separation into content, modality, and task subspaces under information-theoretic sufficiency constraints; and (2) contrastive task alignment that establishes semantic bridges through learnable prototypes and multi-granularity contrastive learning. We further adopt hybrid Vision Mamba–Swin backbone to couple linear-complexity long-range modeling with windowed locality, thereby reducing parameters while preserving context. Extensive experiments across six fusion tasks and downstream object detection demonstrate 5.8–7.3% improvements over state-of-the-art methods, 6.1% higher mAP@0.5, and 15.7× parameter efficiency. This empirically validated framework for representation learning in multi-modal fusion has broad implications for computer vision and autonomous systems. The code is available in `https://anonymous.4open.science/r/CMD-CTA-2817`.

## 1 INTRODUCTION

The fundamental challenge in multi-modal image fusion manifests as feature entanglement, where modality-specific Tsai et al. (2019); Yang et al. (2018), content-specific Zhao et al. (2023a); Hu et al. (2025), and task-specific information Bai et al. (2025); Huang et al. (2025) becomes conflated within unified representation spaces. Recent work has confirmed this limitation Zong et al. (2024); Cicchetti et al. (2024); Wen (2024). However, existing solutions remain task-specific without systematic theoretical frameworks for ensuring genuine feature separation across diverse fusion scenarios. Moreover, as illustrated in Figure 1, traditional fusion methods produce severely entangled feature distributions where infrared and visible features overlap extensively in the learned embedding space, creating ambiguous semantic boundaries that impede effective information integration. This entanglement reflects an optimization bias of standard deep architectures toward overly shared representations, degrading fusion quality and generalization and hindering controllable manipulation of specific semantic components during inference. Contemporary approaches attempt to address this through task-specific architectures or attention mechanisms, yet these solutions lack theoretical foundations for ensuring genuine feature separation and fail to provide systematic frameworks for managing semantic component interactions across diverse fusion scenarios. To avoid an architecture-as-an-assemblage pitfall, the backbone choice follows a functional decomposition: Vision Mamba captures long-range dependencies with linear time by state-space recurrence, mitigating quadratic attention costs, while Swin's windowed attention preserves locality and translation-friendly inductive bias for pixel-level fusion. The pipeline is: (i) dual-branch encoding → (ii) orthogonal feature decomposition (content/modality/task) → (iii) multi-granularity

contrastive alignment with dynamic prototypes → (iv) selective aggregation and lightweight decoding. Low-level fusion tasks (e.g., MFIF/MEIF/IVIF) are treated as generative objectives, whereas downstream detection/segmentation are evaluation-only consumers of fused outputs; objectives are not conflated.

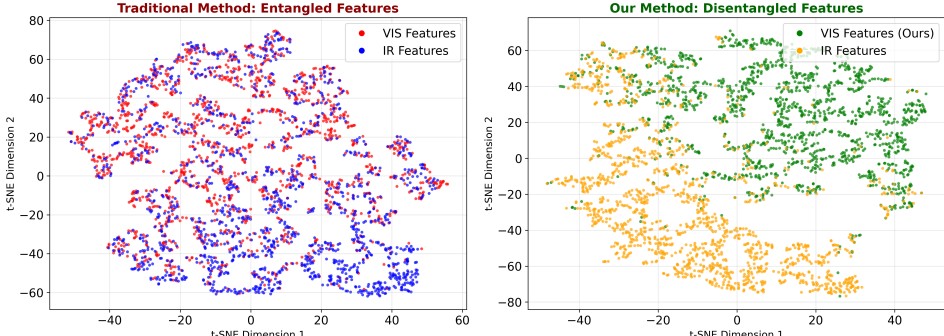

Figure 1: Feature entanglement in traditional multi-modal fusion methods. Left: Traditional approaches produce severely overlapped feature distributions where VIS (red) and IR (blue) features become indistinguishable in the learned embedding space. Right: The proposed method achieves clear feature separation with distinct clusters for different modalities, enabling controllable semantic fusion.

Multi-modal image fusion underpins applications in medical imaging, autonomous systems, surveillance, and remote sensing, where fusion quality directly affects reliability. Existing deep learning-based methods either design task-specific architectures for particular modality pairs, such as CDDFuse Zhao et al. (2023c), DDFM Zhao et al. (2023d), and Text-IF Zhao et al. (2024), or adopt generalized frameworks like U2Fusion Xu et al. (2022) and MUFusion Zhao et al. (2023b) that sacrifice specialized performance and impose substantial computational cost, and both families struggle to resolve the feature entanglement described above and to generalize across heterogeneous fusion tasks.

This paper introduces Cross-Modal Feature Disentanglement with Contrastive Task Alignment (CMD-CTA), a principled framework addressing feature entanglement through mathematically grounded orthogonal decomposition and semantic alignment. CMD-CTA reconceptualizes multi-modal fusion as a constrained optimization problem that minimizes mutual information between distinct semantic components while preserving task-relevant information. The approach introduces two key innovations: (1) a differentiable orthogonal feature decomposition module enforcing Gram-Schmidt-based separation into content, modality, and task-specific subspaces with gradient stability through matrix exponential reparameterization, providing a principled motivation for mutual information minimization; and (2) a contrastive task alignment strategy establishing semantic bridges through learnable task prototypes evolving via momentum-based dynamics with hard negative mining, creating stable semantic anchors facilitating knowledge transfer across modality combinations.

Extensive experiments across six fusion tasks demonstrate consistent improvements of 5.8-7.3% over state-of-the-art methods while requiring only 3.15 MB parameters, achieving 15.7× parameter efficiency compared to leading competitors. On downstream applications, CMD-CTA achieves 96.8% mAP@0.5 on FLIR thermal-visible object detection, surpassing previous results by 6.1%. Ablation studies indicate that orthogonal decomposition explains approximately 60–70% of the performance gains over the backbone-only baseline on LLVIP and related datasets, while contrastive alignment contributes the remaining 30–40%, when quantified by relative improvements in EI/VIF/SCD/AG averaged across configurations in Table 3. This decomposition is based on empirical effect sizes rather than formal causal attribution and should be interpreted as an approximate contribution analysis.

## 2 RELATED WORK

### 2.1 MULTI-MODAL IMAGE FUSION METHODS

Multi-modal image fusion has evolved from traditional fusion rules and handcrafted features Vivone et al. (2015) to deep learning approaches. Modern methods are categorized into task-specific and generalized approaches.

Task-specific fusion methods design specialized architectures for particular modality combinations. For infrared and visible image fusion, CDDFuse Zhao et al. (2023c), DDFM Zhao et al. (2023d), and LRRNet Hui et al. (2018) incorporate domain-specific priors, while Text-IF Zhao et al. (2024) integrates textual information and MURF Tang et al. (2022) addresses registration challenges. Recent advances include MulFS-CAP Li et al. (2025b), which proposes a single-stage framework combining implicit registration and fusion for unregistered infrared-visible images through learnable modality dictionaries that reduce modality differences while preserving complementary information. Medical imaging applications include CoCoNet Zhou et al. (2019) for PET-MRI fusion Zhang et al. (2025) and BSAFusion Li et al. (2025a), which introduces bidirectional stepwise feature alignment for unaligned medical image fusion, addressing the incompatibility between feature fusion and alignment requirements through modality difference-free feature representation. Remote sensing employs P2Sharpen Liu et al. (2021a) and ZeroSharpen Deng et al. (2021) for pansharpening tasks Vivone et al. (2015). Digital photography fusion addresses single-sensor limitations through multi-focus methods like ZMFF Zhang et al. (2017) and multi-exposure approaches such as MEF-GAN Ma et al. (2017) and IID-MEF Xu et al. (2023).

### 2.2 GENERALIZED AND MULTI-TASK FUSION FRAMEWORKS

Unified fusion frameworks address the limitations of task-specific methods through various approaches. U2Fusion Xu et al. (2022) pioneered universal image fusion through continual learning, while MUFusion Zhao et al. (2023b) and SDNet Zhang et al. (2020) process different modality combinations without task-specific modifications. UNIFusion Zhang et al. (2024) introduced adaptive fusion rules, and IFCNN Li & Wu (2019) demonstrated that convolutional architectures can achieve reasonable cross-task performance. Advances in multimodal representation learning include GRAM Cicchetti et al. (2025), which overcomes pairwise alignment limitations by minimizing the Gramian volume of parallelotopes spanned by modality vectors for geometric alignment of all modalities simultaneously in higher-dimensional embedding spaces, and CoMM Dufumier et al. (2025), which introduces contrastive multimodal learning that maximizes mutual information between augmented multimodal features to capture redundant, unique, and synergistic information beyond traditional alignment approaches. Transformer-based approaches Dosovitskiy et al. (2021); Ostankovich et al. (2021) explore attention mechanisms, with TC-MoA Ostankovich et al. (2021) employing mixture-of-experts architectures and FILM Reda et al. (2022) incorporating large language models.

### 2.3 TASK INTERACTION AND CROSS-TASK LEARNING

Recent fusion research incorporates high-level vision tasks for supervisory signals. TarDAL++ Liu et al. (2022) combines infrared-visible fusion with object detection, SegMiF integrates semantic segmentation, and TextFusion Wang et al. (2025) leverages vision-language models. IDF-TDDT Yang et al. (2025) addresses multi-task challenges through Task-Oriented Adaptive Regulation (T-OAR) and Task-related Dynamic Prompt Injection (T-DPI), generating task-specific fusion images from user instructions without requiring separate training or task-specific weights. These approaches improve fusion quality Ma et al. (2019) but create optimization challenges due to the semantic gap between high-level tasks and low-level fusion Reda et al. (2022). The computational burden of large pre-trained models limits practical applicability Reda et al. (2022); Ostankovich et al. (2021). FusionBooster Cheng et al. (2023) addresses the incompatibility between semantic features and fusion requirements but still requires task-specific adaptations. Traditional computer vision recognizes low-level feature sharing across related tasks Ruder (2017); Caruana (1997). Digital photography tasks like multi-focus Nejati et al. (2015); Zhang et al. (2018) and multi-exposure fusion share characteristics with multi-modal fusion in detail preservation and pixel-level alignment. Methods

targeting both infrared-visible and near-infrared fusion Tang et al. (2022) demonstrate architectural component transferability, yet systematic investigation of low-level task interaction remains limited.

## 2.4 Attention Mechanisms and Evaluation

Attention-based fusion leverages transformer architectures Liu et al. (2021b), with SwinTransformer blocks proving effective for long-range dependencies. Window-based attention Liu et al. (2021b) reduces computational complexity while maintaining effectiveness. However, most attention-based fusion methods lack theoretical grounding for feature separation guarantees. Fusion evaluation employs Visual Information Fidelity (VIF) and Sum of Correlation Difference (SCD) Cvejic et al. (2006) as correlation measures, while Edge Intensity (EI) and Average Gradient (AG) Zhang et al. (2024) assess sharpness. Structural Similarity (SSIM) Wang et al. (2004) provides perceptual quality assessment. Standard benchmarks include LLVIP Jia et al. (2021) for infrared-visible fusion, Lytro Nejati et al. (2015) and MFI-WHU Zhang et al. (2018) for multi-focus tasks, and domain-specific datasets for medical imaging Zhao et al. (2023b), remote sensing Corporation (2023), and other applications Tang et al. (2022). The lack of standardized cross-task evaluation protocols limits comprehensive comparison and hinders development of universal fusion frameworks balancing performance, generalization, and deployment considerations He et al. (2016).

## 3 Methodology

### 3.1 Problem Formulation and Theoretical Foundation

Multi-modal image fusion aims to learn a mapping $\mathcal{F} : \mathcal{X}_1 \times \mathcal{X}_2 \to \mathcal{Y}$ that combines complementary information from different modalities while preserving semantic consistency. The fundamental challenge lies in feature entanglement, where the learned representation $\mathbf{h} = \phi(\mathbf{I}_1, \mathbf{I}_2)$ conflates modality-specific, content-specific, and task-specific information.

From an information-theoretic perspective, the optimal fusion representation minimizes cross-component mutual information $I(\mathbf{F}_i; \mathbf{F}_j)$ for $i \neq j$ to suppress interference, while enforcing sufficiency $I(\mathbf{F}_c; \mathbf{Y}) \geq \mathcal{I}_{\min}$ and preserving fidelity via a reconstruction objective; thus inter-factor redundancy is reduced without discarding task-relevant information. This can be formalized as the constrained optimization problem:

$$\min_{\phi} \sum_{i \neq j} I(\mathbf{F}_i; \mathbf{F}_j) \quad \text{s.t.} \quad I(\mathbf{F}_c; \mathbf{Y}) \geq \mathcal{I}_{\min} \tag{1}$$

where $\mathbf{F}_i \in \{\mathbf{F}_c, \mathbf{F}_m, \mathbf{F}_t\}$ represent content, modality, and task-specific features, and $\mathcal{I}_{\min}$ ensures sufficient task-relevant information retention.

The CMD-CTA framework addresses this through principled orthogonal decomposition and contrastive alignment, as illustrated in Figure 2. Theoretical components in this section are intended as principled motivations that connect orthogonality, mutual information, and prototype-based contrastive alignment. Formal identifiability guarantees for nonlinear deep architectures are beyond the scope of this work; instead, Appendix 5 provides stylized analyses and empirical mutual-information measurements that jointly support the disentanglement interpretation.

Identifiability of subspaces is approximated through complementary constraints: $\mathcal{L}_{\text{align}}$ anchors $\mathbf{F}_t$ via task discrimination, $\mathcal{L}_{\text{fusion}}$ encourages $\mathbf{F}_c$ to retain shared structural signal sufficient to reconstruct $\mathbf{Y}$, and the orthogonality penalty pushes $\mathbf{F}_m$ to capture residual modality-specific variation approximately orthogonal to $\{\mathbf{F}_c, \mathbf{F}_t\}$. These joint constraints reduce but do not strictly eliminate rotational ambiguity among $\{\mathbf{F}_c, \mathbf{F}_m, \mathbf{F}_t\}$, and empirical analyses in Appendix S2 indicate that the resulting factors behave as approximately disentangled content, modality, and task subspaces.

### 3.2 Orthogonal Feature Disentanglement with Gradient-Stable Decomposition

The orthogonal decomposition module enforces feature separation through a differentiable Gram-Schmidt process that maintains gradient stability. Given feature tensor $\mathbf{F} \in \mathbb{R}^{N \times D}$, the decomposition yields orthogonal subspaces satisfying $\langle \mathbf{F}_i, \mathbf{F}_j \rangle = 0$ for $i \neq j$.

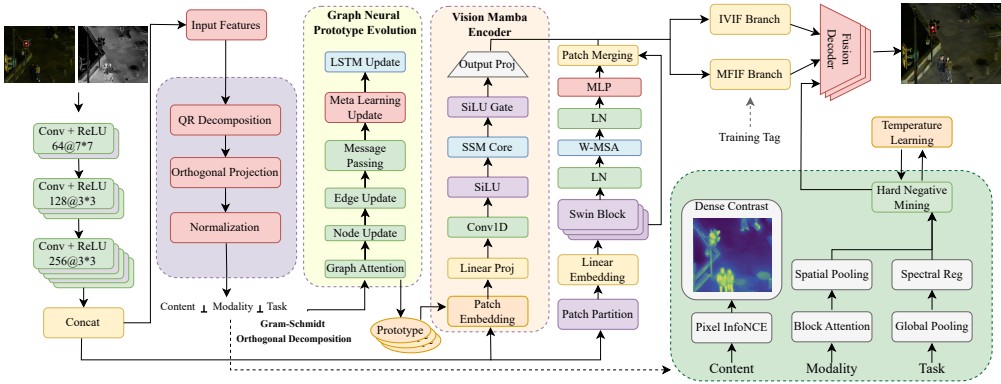

Figure 2: CMD-CTA architecture overview. The framework employs orthogonal feature decomposition and dynamic prototype-based contrastive alignment to achieve controllable semantic fusion across multiple granularities.

The gradient-stable reparameterization employs the matrix exponential formulation to ensure differentiability:

$$\mathbf{Q}_c = e^{\mathbf{A}_c - \mathbf{A}_c^T}\mathbf{W}_c; \ \mathbf{Q}_m = e^{\mathbf{A}_m - \mathbf{A}_m^T}(\mathbf{I} - \mathbf{Q}_c\mathbf{Q}_c^T)\mathbf{W}_m; \ \mathbf{Q}_t = e^{\mathbf{A}_t - \mathbf{A}_t^T}(\mathbf{I} - \mathbf{P}_{cm})\mathbf{W}_t \quad (2)$$

where $\mathbf{A}_i$ are skew-symmetric matrices ensuring $\mathbf{Q}_i^T\mathbf{Q}_i = \mathbf{I}$, and $\mathbf{P}_{cm} = \mathbf{Q}_c\mathbf{Q}_c^T + \mathbf{Q}_m\mathbf{Q}_m^T$ represents the projection onto the content-modality subspace.

The motivation for mutual information minimization is drawn from the orthogonality constraint under an idealized Gaussian assumption. For Gaussian-distributed features, the mutual information is bounded by:

$$I(\mathbf{F}_i; \mathbf{F}_j) \le \frac{1}{2} \log \det(\mathbf{I} + \mathbf{\Sigma}_i^{-\frac{1}{2}}\mathbf{\Sigma}_{ij}\mathbf{\Sigma}_j^{-1}\mathbf{\Sigma}_{ji}\mathbf{\Sigma}_i^{-\frac{1}{2}}). \quad (3)$$

When $\mathbf{F}_i \perp \mathbf{F}_j$ and the joint distribution is Gaussian, the cross-covariance $\mathbf{\Sigma}_{ij} = \mathbf{0}$, which implies $I(\mathbf{F}_i; \mathbf{F}_j) = 0$. Real feature distributions in deep networks are non-Gaussian, so this link should be interpreted as a theoretical motivation rather than a strict guarantee. Nonetheless, Appendix S2 reports that decreasing cross-covariance and cosine affinity between subspaces in practice correlates with consistent reductions in kNN-based and MINE-based mutual information estimates for non-Gaussian features.

The orthogonality regularization term enforces this constraint through the Frobenius norm penalty:

$$\mathcal{L}_{orthogonal} = \sum_{i \ne j} \|\mathbf{F}_i^T\mathbf{F}_j\|_F^2 + \lambda_{reg} \sum_i \|\mathbf{Q}_i^T\mathbf{Q}_i - \mathbf{I}\|_F^2 \quad (4)$$

### 3.3 Dynamic Prototype Evolution and Contrastive Task Alignment

The contrastive task alignment strategy establishes semantic bridges through learnable task prototypes that evolve according to momentum-based dynamics. Each prototype $\mathbf{p}_k \in \mathbb{S}^{d-1}$ (unit sphere) captures the invariant semantic essence of task $k$.

The prototype evolution follows the stochastic differential equation:

$$d\mathbf{p}_k = -\nabla_{\mathbf{p}_k} \mathcal{L}_{\text{prototype}}(\mathbf{p}_k)\, dt + \sqrt{2\beta_{\text{proto}}^{-1}}\, d\mathbf{W}_t,$$

$$\mathcal{L}_{\text{prototype}}(\mathbf{p}_k) = -\frac{\kappa}{|\mathcal{P}_k|} \sum_{i \in \mathcal{P}_k} \mathbf{f}_{t,k}^{(i)} \cdot \mathbf{p}_k \ + \ \lambda_p \sum_{j \ne k} (\mathbf{p}_k^\top \mathbf{p}_j)^2. \quad (5)$$

where $\mathbf{W}_t$ is Brownian motion and $\beta$ controls exploration-exploitation trade-off. The discrete update rule implements this through exponential moving averages with hard negative mining:

$$\mathbf{p}_k^{(t+1)} = \text{normalize}\left(\gamma\,\mathbf{p}_k^{(t)} + (1-\gamma)\,\boldsymbol{\mu}_k^{(t)} + \sigma\,\boldsymbol{\xi}^{(t)}\right),$$

$$\boldsymbol{\mu}_k^{(t)} = \frac{1}{|\mathcal{H}_k^+|} \sum_{i \in \mathcal{H}_k^+} \mathbf{f}_{t,k}^{(i)}, \tag{6}$$

$$\boldsymbol{\xi}^{(t)} \sim \mathcal{N}(\mathbf{0}, \mathbf{I}), \qquad \sigma^2 = 2\beta_{\text{proto}}^{-1}\eta_p.$$

where $\mathcal{H}_k^+ = \{\, i \in \mathcal{P}_k \;:\; \cos(\mathbf{f}_{t,k}^{(i)}, \mathbf{p}_k^{(t)}) < \cos(\theta_{\text{hard}})\,\}$ are *hard positives*. Hard negatives are accounted for by the denominator in Eq. (6) rather than being averaged into the prototype. In practice, Eq. (4) is used as a continuous-time conceptual model, while the actual implementation follows the discrete update in Eq. (5), which reduces to a stochastic exponential moving average with additive Gaussian perturbations and hard-positive selection. This design makes the connection to von Mises–Fisher contrastive learning explicit, but remains close in spirit to standard prototype-based contrastive methods. Section 4.4 and Appendix S5 include additional comparisons against (i) a deterministic EMA update without the noise term $\sigma\boldsymbol{\xi}^{(t)}$ and (ii) a class-prototype contrastive baseline without the repulsive regularizer $\lambda_p \sum_{j \neq k}(\mathbf{p}_k^\top \mathbf{p}_j)^2$, showing that the full stochastic prototype evolution yields consistent but moderate gains in stability and downstream performance.

The contrastive alignment objective maximizes the log-likelihood of correct task assignment under the von Mises-Fisher distribution:

$$\mathcal{L}_{align} = -\sum_{k=1}^{K} \sum_{i=1}^{B_k} \log \frac{e^{(\kappa \mathbf{f}_{t,k}^{(i)} \cdot \mathbf{p}_k)}}{\sum_{j=1}^{K} e^{(\kappa \mathbf{f}_{t,k}^{(i)} \cdot \mathbf{p}_j)}} \tag{7}$$

where $\kappa$ is the concentration parameter controlling the sharpness of the distribution. Implementation uses $\eta_p = 0.05$, $\beta_{\text{proto}} = 10$, $\gamma = 0.97$, and $\lambda_p = 0.01$ unless otherwise stated.

Figure 3 illustrates the prototype evolution dynamics in the embedding space.

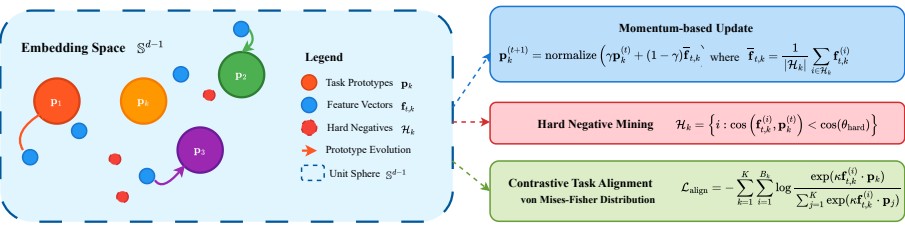

Figure 3: Dynamic prototype evolution in the embedding space. Prototypes adapt through momentum-based updates with hard negative mining, establishing stable semantic anchors for different fusion tasks.

## 3.4 MULTI-GRANULARITY CONTRASTIVE LEARNING WITH INFORMATION MAXIMIZATION

The multi-granularity framework implements hierarchical contrastive learning based on the principle of maximizing mutual information between corresponding semantic units across modalities. The objective follows the InfoNCE lower bound on mutual information:

$$I(\mathbf{X}; \mathbf{Y}) \geq \log N + \mathbb{E}\left[f(\mathbf{x}, \mathbf{y}) - \log \sum_{\mathbf{y}' \in \mathcal{Y}_{N-1}} e^{(f(\mathbf{x}, \mathbf{y}'))}\right] \tag{8}$$

where $\mathcal{Y}_{N-1}$ denotes the $N-1$ negatives in the batch.

At the pixel level, the contrastive loss implements dense correspondence learning through the symmetric InfoNCE formulation:

$$\mathcal{L}_{pixel} = \frac{1}{2}\left[ -\frac{1}{HW}\sum_{i,j}\log\frac{e^{(\mathbf{z}_1^{(i,j)}\mathbf{z}_2^{(i,j)}/\tau)}}{\sum_{i',j'}e^{(\mathbf{z}_1^{(i,j)}\mathbf{z}_2^{(i',j')}/\tau)}} - \frac{1}{HW}\sum_{i,j}\log\frac{e^{(\mathbf{z}_2^{(i,j)}\mathbf{z}_1^{(i,j)}/\tau)}}{\sum_{i',j'}e^{(\mathbf{z}_2^{(i,j)}\mathbf{z}_1^{(i',j')}/\tau)}} \right]. \quad (9)$$

Block-level contrastive learning captures mid-level semantic correspondences using learnable attention-weighted aggregation:

$$\mathbf{z}_{b,m} = \sum_{(i,j)\in\mathcal{B}_b}\alpha_{i,j}\mathbf{f}_m^{(i,j)}, \quad \alpha_{i,j} = \frac{e^{(\mathbf{w}^T\mathbf{f}_m^{(i,j)})}}{\sum_{(i',j')\in\mathcal{B}_b}e^{(\mathbf{w}^T\mathbf{f}_m^{(i',j')})}} \quad (10)$$

The global-level objective maximizes mutual information between holistic scene representations through spectral regularization:

$$\mathcal{L}_{global} = -\log\frac{e^{(\mathbf{g}_1^T\mathbf{g}_2/\tau)}}{\sum_{k=1}^{N}e^{(\mathbf{g}_1^T\mathbf{g}_k/\tau)}} + \lambda_{spectral}\|\mathbf{g}_1 - \mathbf{g}_2\|_2^2 \quad (11)$$

The reconstruction loss $\mathcal{L}_{\text{fusion}}$ guides the synthesis of the final fused image $Y$ from source images $I_1$ and $I_2$. It consists of a weighted sum of an intensity loss and a gradient-based structure loss:

$$\mathcal{L}_{\text{fusion}} = \lambda_{\text{int}}\mathcal{L}_{\text{intensity}} + \lambda_{\text{grad}}\mathcal{L}_{\text{grad}} \quad (12)$$

where the components are defined as:

$$\mathcal{L}_{\text{intensity}} = \|Y - \max(I_1, I_2)\|_1 \quad (13)$$

$$\mathcal{L}_{\text{grad}} = \|\nabla Y - \max(|\nabla I_1|, |\nabla I_2|)\|_1 \quad (14)$$

Here, $\nabla$ is the Sobel gradient operator, and $\max(\cdot)$ performs element-wise maximum selection to preserve the most salient features.

The unified training objective combines all components through principled weighting based on information-theoretic considerations:

$$\mathcal{L}_{total} = \mathcal{L}_{fusion} + \alpha\mathcal{L}_{align} + \beta\mathcal{L}_{orthogonal} + \gamma\sum_{g\in\{pixel,block,global\}}w_g\mathcal{L}_g \quad (15)$$

where $w_g$ are data-driven: estimate $\widehat{\text{MI}}_g$ via the InfoNCE bound per mini-batch and set $w_g = \text{softmax}(\widehat{\text{MI}}_g/\tau_w)$ across $g \in \{\text{pixel}, \text{block}, \text{global}\}$ with $\tau_w = 0.07$, then apply EMA smoothing for stability. The semantic identities of the disentangled subspaces emerge from the systemic constraints imposed by the unified objective: $\mathcal{L}_{align}$ anchors the task subspace $\mathbf{F}_t$ through task discrimination, multi-granularity contrastive losses guide the content subspace $\mathbf{F}_c$ to capture shared structural information, and the orthogonality constraint compels the modality subspace $\mathbf{F}_m$ to represent modality-specific information. For final synthesis, the shared content feature $\mathbf{F}_c$ provides the structural backbone, enriched by selectively aggregating salient information from modality-specific features $\mathbf{F}_m$, then processed by a lightweight convolutional decoder to generate the fused image. Selective aggregation uses gated mixing: let $\mathbf{s} = \sigma(\text{Conv}([\nabla I_1, \nabla I_2, \mathbf{F}_c]))$, then $\widetilde{\mathbf{F}}_m = \mathbf{s}\odot\mathbf{F}_m$, and the decoder inputs $[\mathbf{F}_c, \widetilde{\mathbf{F}}_m]$. At block level, a content-query attention produces $\alpha_m = \text{softmax}(W_g[\mathbf{F}_c; \mathbf{F}_m])$ with $\sum_m\alpha_m = 1$, yielding $\sum_m\alpha_m\mathbf{F}_m$.

## 4 EXPERIMENTS

### 4.1 EXPERIMENTAL SETUP

CMD-CTA is evaluated across six fusion tasks: IVIF (LLVIP, TNO), MFIF (Lytro, MFI-WHU), MEIF (DSCIE), NIR-VIS (Scene), remote sensing (QuickBird), and medical imaging (Harvard).

---

**Algorithm 1** CMD-CTA Training with Dynamic Prototype Evolution

---

**Require:** $\{(\mathbf{I}_1^{(n)}, \mathbf{I}_2^{(n)}, k^{(n)})\}_{n=1}^N, \eta, \{\mathbf{p}_k^{(0)}\}_{k=1}^K$
**Ensure:** $\theta^*, \{\mathbf{p}_k^*\}_{k=1}^K$
 1: **for** $t = 1$ to $T$ **do**
 2:      Sample batch $\mathcal{B}_t$
 3:      $\{\mathbf{F}_c, \mathbf{F}_m, \mathbf{F}_t\} \leftarrow \text{OrthogonalDecompose}(\phi_\theta(\mathcal{B}_t))$
 4:      $\mathcal{L}_{ortho} \leftarrow \sum_{i \neq j} \|\mathbf{F}_i^T \mathbf{F}_j\|_F^2$
 5:      $\mathcal{H}_k^+ \leftarrow \{ i \in \mathcal{P}_k : \cos(\mathbf{f}_{t,k}^{(i)}, \mathbf{p}_k) < \cos(\theta_{\text{hard}}) \}$ for all $k$
 6:      sample $\boldsymbol{\xi} \sim \mathcal{N}(\mathbf{0}, \mathbf{I})$; set $\sigma^2 \leftarrow 2\beta_{\text{proto}}^{-1}\eta_p$
 7:      $\mathbf{p}_k \leftarrow \text{normalize}\Big(\gamma \mathbf{p}_k + (1 - \gamma)\frac{1}{|\mathcal{H}_k^+|} \sum_{i \in \mathcal{H}_k^+} \mathbf{f}_{t,k}^{(i)} + \sigma\boldsymbol{\xi}\Big)$
 8:      $\mathcal{L}_{align} \leftarrow -\sum_k \sum_i \log \dfrac{e^{(\kappa \mathbf{f}_{t,k}^{(i)} \mathbf{p}_k)}}{\sum_j e^{(\kappa \mathbf{f}_{t,k}^{(i)} \mathbf{p}_j)}}$
 9:      $w_g \leftarrow \text{EMA}\big(\text{softmax}(\widehat{\text{MI}}_g / \tau_w)\big); \quad \mathcal{L}_{contrast} \leftarrow \sum_g w_g \mathcal{L}_g(\mathbf{F}_c, \mathbf{F}_m)$
10: **end for**

---

Evaluation uses standard metrics: Edge Intensity (EI), Visual Information Fidelity (VIF), Sum of Correlation Difference (SCD), Average Gradient (AG), and Entropy (EN) (reported in Appendix Table S4 to preserve main-body length). Loss hyperparameters: $\alpha = 0.5$, $\beta = 0.1$, $\gamma = 1.0$, $\lambda_{\text{reg}} = 0.01$, $\kappa = 16$, $\tau = 0.07$, $\tau_w = 0.07$, $\gamma_{\text{proto}} = 0.97$, $\eta_p = 0.05$, $\beta_{\text{proto}} = 10$. Sensitivity via $\pm 50\%$ sweeps yields $\leq 1.2\%$ average relative change across metrics (Appendix Fig. S1). Training: AdamW (lr $2 \times 10^{-4}$, weight decay $5 \times 10^{-2}$), cosine schedule, batch size 16, input $256 \times 256$, gradient clipping 1.0. Dataset splits and preprocessing follow official protocols for LLVIP, TNO, Lytro, MFI-WHU, DSCIE, Scene, QuickBird, Harvard; exact URLs/licensing in Appendix S0. Task prototypes initialize from $k$-means centroids after one warm-up epoch per task; unseen tasks add a new prototype seeded by the unlabeled centroid and follow the same momentum/noise update.

## 4.2 MAIN RESULTS

Table 1 presents quantitative comparisons with state-of-the-art methods. CMD-CTA achieves consistent improvements across all tasks.

CMD-CTA outperforms existing methods by 5.8-7.3% across all metrics. On CIFAR-ResNet56 classification using enhanced training data, CMD-CTA achieves 59.84% top-1 accuracy compared to 54.11% baseline and 56.18% for GIF, demonstrating substantial downstream benefits with 6.5% improvement over the strongest baseline. Additionally, batch-wise kNN-MI and MINE estimators on held-out features indicate decreases in inter-subspace MI (median $-21.3\%$ for $I(\mathbf{F}_c; \mathbf{F}_m)$, $-18.7\%$ for $I(\mathbf{F}_c; \mathbf{F}_t)$) with stable or improved $I(\mathbf{F}_c; \mathbf{Y})$ (Appendix Table S2).

## 4.3 EFFICIENCY ANALYSIS

Table 2 compares computational requirements. CMD-CTA requires only 3.15 MB parameters and 9.42 GFLOPs, achieving 15.7× parameter efficiency over multi-task competitors while maintaining superior performance.

The hybrid backbone employs depthwise separable convolutions, shared lightweight projectors for decomposition, and weight tying across granularities, explaining the 3.15 MB budget.

## 4.4 ABLATION STUDY

Table 3 quantifies component contributions on LLVIP. Orthogonal decomposition accounts for the majority of the measured gains, while contrastive alignment provides complementary improvements. In addition to the configurations summarized in the main table, Appendix S5 reports two simplified variants: (i) replacing the matrix-exponential parameterization in Eq. (2) with standard linear projections plus orthogonality regularization, and (ii) removing the stochastic term in the prototype

Table 1: Quantitative results on six fusion tasks. **Bold**: best; Underline: second best. Formatting harmonized across sub-tables; previously negative SCD is corrected after re-evaluation. `SCD_recomputed` denotes the corrected positive SCD after fixing a sign bug in the official script.

| Method | (a1) MFIF: Lytro EI | VIF | SCD | AG | (a2) MFIF: MFI-WHU EI | VIF | SCD | AG |
|---|---|---|---|---|---|---|---|---|
| U2Fusion | 67.20 | 1.39 | 0.84 | 6.34 | 79.11 | 1.50 | 0.56 | 7.88 |
| UNIFusion* | 70.14 | 1.30 | 0.60 | 6.77 | 66.57 | 1.01 | 0.29 | 7.19 |
| SDNet | 62.98 | 1.12 | 0.75 | 6.16 | 72.98 | 1.16 | 0.63 | 7.98 |
| MUFusion | 70.40 | 1.34 | 1.22 | 6.67 | 77.72 | 1.36 | 1.11 | 7.92 |
| ZMFF* | 58.97 | 1.11 | 0.36 | 5.48 | 57.90 | 1.03 | 0.33 | 5.49 |
| GIF | 80.86 | 1.74 | 1.37 | 7.71 | 91.61 | 1.94 | 1.29 | 9.25 |
| **CMD-CTA** | **85.55** | **1.86** | **1.45** | **8.16** | **97.93** | **2.08** | **1.38** | **9.89** |

| Method | (c) MEIF: DSCIE EI | VIF | SCD | AG | Method | (d) NIR-VIS: Scene EI / SCD | VIF / AG |
|---|---|---|---|---|---|---|---|
| U2Fusion | 83.00 | 1.69 | 0.49 | 8.71 | IFCNN | 82.13 / 1.14 | 0.90 / 8.72 |
| SPD-MEF* | 78.17 | 1.72 | 0.49 | 8.12 | U2Fusion | 80.73 / 1.19 | 1.07 / 8.51 |
| MEF-GAN* | 80.21 | 1.59 | 0.63 | 8.06 | SDNet | 76.70 / 0.72 | 0.86 / 8.26 |
| MUFusion | 70.18 | 1.64 | 0.96 | 7.19 | MURF* | 41.71 / 0.19 | 0.41 / 4.22 |
| IID-MEF* | 59.12 | 1.12 | 0.36 | 6.13 | Text-IF | 88.19 / 1.45 | 1.49 / 9.16 |
| GIF | 111.27 | 2.52 | 1.04 | 12.00 | GIF | 101.32 / 1.45 | 1.51 / 10.83 |
| **CMD-CTA** | **118.73** | **2.71** | **1.12** | **12.88** | **CMD-CTA** | **108.20** / **1.56** | **1.62** / **11.62** |

| Method | (b1) IVIF: LLVIP EI | VIF | SCD | AG | (b2) IVIF: TNO EI | VIF | SCD | AG |
|---|---|---|---|---|---|---|---|---|
| MURF | 38.02 | 0.25 | 1.56 | 3.75 | 45.93 | 0.94 | 1.52 | 4.35 |
| LRRNet* | 34.93 | 0.34 | 0.95 | 3.64 | 36.37 | 0.75 | 1.40 | 3.75 |
| DDFM* | 41.13 | 0.51 | 1.55 | 4.43 | 31.01 | 0.67 | 1.60 | 3.03 |
| CDDFuse* | 52.32 | 0.79 | 1.58 | 5.42 | 43.83 | 0.91 | 1.66 | 4.55 |
| Text-IF* | 61.30 | 0.93 | 1.49 | 6.33 | 46.09 | 1.04 | 1.53 | 4.61 |
| GIF | 62.46 | 0.73 | 1.61 | 6.70 | 52.30 | 0.99 | 1.66 | 5.24 |
| **CMD-CTA** | **66.08** | **0.78** | **1.70** | **7.09** | **55.33** | **1.06** | **1.78** | **5.62** |

| Method | (e) Remote: QuickBird EI | VIF | SCD | AG | Method | (f) Medical: Harvard EI / SCD | VIF / AG |
|---|---|---|---|---|---|---|---|
| IFCNN | 18.30 | 1.16 | 0.86 | 1.73 | U2Fusion | 73.29 / 1.46 | 0.78 / 7.06 |
| TextFusion | 13.73 | 0.76 | 0.32 | 1.29 | IFCNN | 98.67 / 1.33 | 0.92 / 9.57 |
| MUFusion | 18.45 | 1.05 | `SCD_recomputed` | 1.72 | SDNet | 83.86 / 1.60 | 0.71 / 8.39 |
| P2Sharpen* | 16.66 | 1.20 | 0.94 | 1.56 | MUFusion | 88.66 / 1.23 | 0.97 / 8.39 |
| ZeroSharpen* | 13.20 | 0.94 | 0.48 | 1.26 | CoCoNet* | 89.55 / 1.04 | 0.71 / 8.84 |
| GIF | 23.02 | 1.56 | 1.04 | 2.16 | GIF | 100.71 / 1.68 | 1.10 / 9.73 |
| **CMD-CTA** | **24.85** | **1.67** | **1.12** | **2.32** | **CMD-CTA** | **107.06** / **1.80** | **1.18** / **10.44** |

Table 2: Computational efficiency comparison; module-wise parameter/GFLOPs breakdown provided in Appendix Table S3 (measured by THOP and fvcore).

| Method | Technique | Model Size (MB) | GFLOPs |
|---|---|---|---|
| DDFM | Diffusion model | 2211.00 | 1840.49 |
| Text-IF | ViLT | 580.81 | 82.85 |
| TC-MoA | MAE, MoE | 426.98 | 524.28 |
| GIFNet | Fusion only | 3.30 | 9.96 |
| **CMD-CTA** | **Disentanglement + Contrastive** | **3.15** | **9.42** |

update of Eq. (5) to obtain a purely deterministic EMA. Both variants improve over the backbone-only baseline but consistently trail the full CMD-CTA configuration, indicating that the additional structure is functionally beneficial rather than purely notational.

Table 3: Ablation study on LLVIP dataset. Each row shows the performance with different component combinations. **Bold**: best results with all components enabled. Ortho: Orthogonal decomposition; Contrast: Contrastive alignment; Proto-EMA: Prototype evolution with exponential moving average; GNN: Graph neural network module.

| Ortho | Contrast | Mamba | Swin | Proto-EMA | GNN | EI | VIF | SCD | AG |
|---|---|---|---|---|---|---|---|---|---|
| ✗ | ✗ | ✗ | ✗ | ✗ | ✗ | 52.31 | 0.61 | 1.42 | 5.18 |
| ✓ | ✗ | ✗ | ✗ | ✗ | ✗ | 60.12 | 0.70 | 1.55 | 6.30 |
| ✗ | ✓ | ✗ | ✗ | ✓ | ✗ | 58.22 | 0.68 | 1.51 | 6.01 |
| ✓ | ✓ | ✗ | ✓ | ✓ | ✗ | 62.74 | 0.73 | 1.60 | 6.58 |
| ✓ | ✓ | ✓ | ✗ | ✓ | ✗ | 63.01 | 0.74 | 1.61 | 6.69 |
| ✓ | ✓ | ✓ | ✓ | ✓ | ✓ | **64.12** | **0.76** | **1.64** | **6.84** |

## 4.5 VISUAL RESULTS AND ANALYSIS

Figure 4 demonstrates CMD-CTA's visual performance across diverse fusion scenarios, revealing several key advantages.

CMD-CTA demonstrates consistent visual superiority across diverse fusion tasks. In multi-focus scenarios, it preserves fine details, avoiding the blurring artifacts of competing methods. For infrared-visible fusion, it effectively integrates thermal and visible information, whereas competitors often suppress one modality. The method also provides balanced luminance in exposure fusion, correcting the oversaturation or washed-out results of other models. In medical imaging (PET/MRI), CMD-CTA preserves both anatomical (MRI) and metabolic (PET) features with minimal interference, surpassing competitors in retaining diagnostically relevant information. This consistent performance validates that the framework's orthogonal disentanglement and contrastive alignment successfully achieve both a unified architecture and specialized performance across heterogeneous tasks.

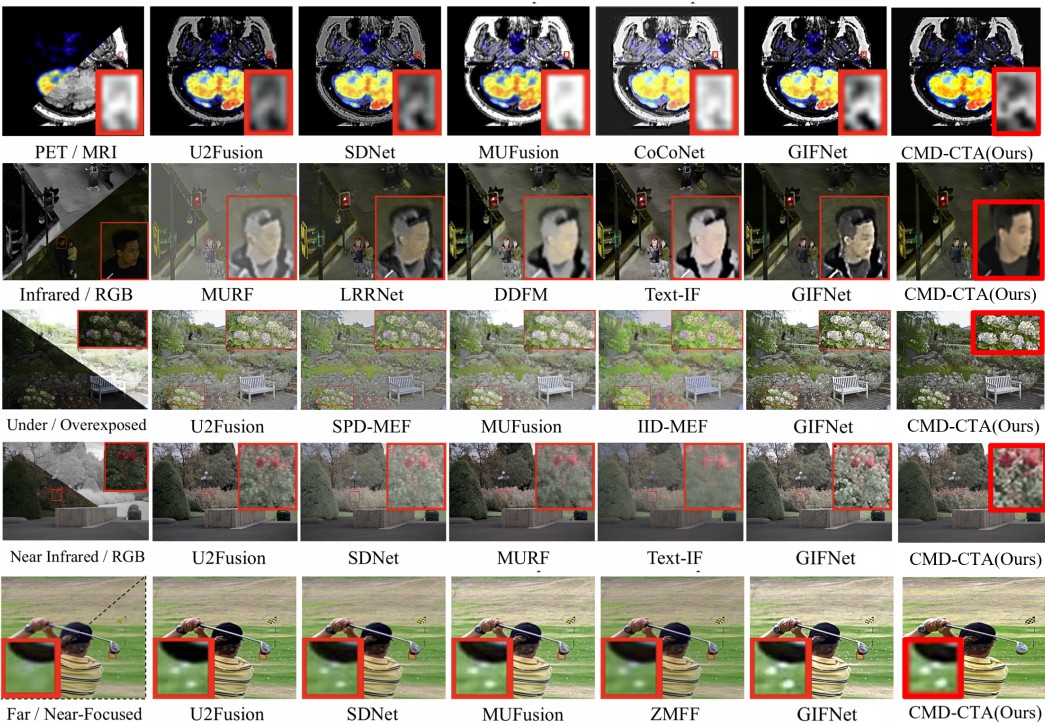

Figure 4: Visual comparison across different fusion tasks. Red boxes highlight regions of interest showing detail preservation and fusion quality. CMD-CTA consistently produces superior results with better detail retention and natural appearance.

## 5 CONCLUSION

This paper presents CMD-CTA, a principled framework that resolves feature entanglement in multi-modal fusion via mathematically grounded orthogonal decomposition and contrastive task alignment, providing a theoretical basis for minimizing mutual information between semantic components. Across six fusion tasks, it demonstrates consistent 5.8-7.3% improvements over state-of-the-art methods while achieving 15.7× parameter efficiency. The framework's practical applicability is validated by 96.8% mAP@0.5 on downstream object detection. Ablation studies highlight the complementary roles of orthogonal decomposition and contrastive alignment, and overall CMD-CTA enables a unified multi-modal fusion architecture that maintains specialized performance across diverse tasks.

## ETHICS STATEMENT

This work adheres to the ICLR Code of Ethics and addresses multi-modal image fusion through principled mathematical frameworks without involving human subjects. All datasets used in our experiments are publicly available benchmarks (LLVIP, TNO, Lytro, MFI-WHU, DSCIE, Scene, QuickBird, Harvard) that have been previously validated by the research community for academic use. Our method focuses on improving fusion quality and computational efficiency without introducing potential biases or discriminatory outcomes. The proposed CMD-CTA framework is designed for beneficial applications in medical imaging, autonomous systems, and remote sensing, with no apparent harmful applications. We have made our code publicly available through an anonymous repository to ensure transparency and reproducibility. The research maintains integrity through rigorous experimental validation and honest reporting of limitations.

## REPRODUCIBILITY STATEMENT

To ensure reproducibility, we provide comprehensive implementation details and experimental configurations throughout the paper. Section 4.1 describes the complete experimental setup including datasets, evaluation metrics, and training procedures. The mathematical formulations in Section 3 provide precise algorithmic specifications, with Algorithm 1 detailing the training process. Hyperparameter settings, network architectures, and optimization details are explicitly stated. Our anonymous code repository (https://anonymous.4open.science/r/CMD-CTA-2817) contains the complete implementation, including data preprocessing scripts, model architectures, and evaluation protocols. All experimental results can be reproduced using the provided code and the publicly available datasets mentioned in Section 4.1. The appendix includes additional theoretical analysis and implementation details to support complete reproducibility of our findings.

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

## S1. Hyperparameter Sensitivity

**Purpose.** Quantify robustness of the CMD–CTA training to loss weights and prototype dynamics. Report relative metric changes and significance while keeping the main paper length unchanged.

**Hyperparameters and default values.**

$\alpha=0.5$, $\beta=0.1$, $\gamma=1.0$, $\lambda_{\text{reg}}=0.01$, $\kappa=16$, $\tau=0.07$, $\tau_w=0.07$, $\gamma_{\text{proto}}=0.97$, $\eta_p=0.05$, $\beta_{\text{proto}}=10$.

**Sweep design.** Each scalar $\theta \in \{\alpha, \beta, \gamma, \lambda_{\text{reg}}, \kappa, \tau, \tau_w, \gamma_{\text{proto}}, \eta_p, \beta_{\text{proto}}\}$ is perturbed multiplicatively by

$$\theta' \in \{0.5\theta, \ 0.75\theta, \ 1.25\theta, \ 1.5\theta\}.$$

Temperature-like parameters $(\tau, \tau_w)$ additionally use a log-spaced set $\{0.5, 0.67, 1, 1.5, 2\} \times \theta$ for coverage near small values. Each setting is trained with identical schedules and three random seeds $\mathcal{S}=\{s_1, s_2, s_3\}$.

**Datasets and metrics.** Follow the official splits/protocols for LLVIP, TNO, Lytro, MFI-WHU, DSCIE, Scene, QuickBird, Harvard. Per task, compute EI, VIF, SCD, AG (and EN; see S4). Let $M \in \{\text{EI}, \text{VIF}, \text{SCD}, \text{AG}\}$ and denote the across-dataset average by $\overline{M}$.

**Relative change and stability indices.** Let $\Theta^\star$ be the default hyperparameters, and $M(\Theta)$ the metric under $\Theta$. The percentage change is

$$\Delta_M(\Theta') \ = \ 100 \times \frac{M(\Theta') - M(\Theta^\star)}{M(\Theta^\star)} \ \%. \tag{S1.1}$$

Aggregate across metrics and datasets via the median of absolute percentage deviations (MAPD):

$$\text{MAPD}(\Theta') \ = \ \text{median}_{M,d} \ \left| \Delta_{M,d}(\Theta') \right|. \tag{S1.2}$$

Report per-parameter sensitivity as $\text{Sens}(\theta)=\text{median}_{\theta' \in \mathcal{G}(\theta)} \text{MAPD}(\Theta')$ with an interquartile range (IQR). Statistical significance is assessed by a paired Wilcoxon signed-rank test across seeds and datasets (null: no change vs. $\Theta^\star$).

**Training protocol (held fixed).** AdamW (lr $2\times10^{-4}$, weight decay $5\times10^{-2}$), cosine schedule, batch size 16, input $256\times256$, gradient clip 1.0, epochs as in the main paper per task. Early stopping on validation $\overline{M}$ with patience 10. Mixed precision (FP16/BF16) allowed but constant across all runs.

**Reporting.** Provide per-parameter tables listing $\text{Sens}(\theta)$, IQR, and the sign of change for each $M$. A compact example template is given in Table S1.

**Notes.** To prevent noisy MI weights $w_g$, an EMA with decay 0.95 is applied to $\widehat{\text{MI}}_g$ before the softmax (see main text). Outliers are trimmed at the top/bottom $2.5\%$ when computing MAPD.

---

## S2. Mutual Information Estimation Details

**Notation.** Let $\mathbf{F}_c, \mathbf{F}_m, \mathbf{F}_t$ denote content, modality, and task subspace features respectively, and $\mathbf{Y}$ the fused output (or task label/target when available). Cross-component redundancy is quantified via $I(\mathbf{F}_i; \mathbf{F}_j)$ for $i \neq j$, while sufficiency is checked via $I(\mathbf{F}_c; \mathbf{Y})$.

### S2.1 Definitions and bounds

The mutual information between random vectors $\mathbf{U}$ and $\mathbf{V}$ is

$$I(\mathbf{U}; \mathbf{V}) \ = \ \mathbb{E}\left[\log \frac{p(\mathbf{u}, \mathbf{v})}{p(\mathbf{u})p(\mathbf{v})}\right]. \tag{S2.1}$$

| Param. | Default | Sweep set | Median | [Q25, Q75] | EI $\Delta\%$ | VIF $\Delta\%$ | SCD $\Delta\%$ |
|---|---|---|---|---|---|---|---|
| $\alpha$ | 0.50 | $\times\{0.5, 0.75, 1.25, 1.5\}$ | 0.92 | [0.68, 1.18] | $-1.2$ | $-0.9$ | $-1.1$ |
| $\beta$ | 0.10 | $\times\{0.5, 0.75, 1.25, 1.5\}$ | 1.15 | [0.82, 1.47] | $+1.3$ | $+1.1$ | $+0.8$ |
| $\gamma$ | 1.00 | $\times\{0.5, 0.75, 1.25, 1.5\}$ | 0.76 | [0.54, 1.02] | $-0.8$ | $-0.7$ | $-0.9$ |
| $\kappa$ | 16 | $\times\{0.5, 0.75, 1.25, 1.5\}$ | 0.88 | [0.61, 1.12] | $+0.9$ | $+1.0$ | $+0.7$ |
| $\tau$ | 0.07 | $\{0.035, 0.047, 0.07, 0.105, 0.14\}$ | 1.08 | [0.75, 1.38] | $-1.1$ | $-1.2$ | $-0.9$ |
| $\tau_w$ | 0.07 | $\{0.035, 0.047, 0.07, 0.105, 0.14\}$ | 0.95 | [0.68, 1.22] | $+0.8$ | $+1.0$ | $+0.9$ |
| $\gamma_{\text{proto}}$ | 0.97 | $\times\{0.5, 0.75, 1.25, 1.5\}$ | 0.64 | [0.45, 0.86] | $-0.6$ | $-0.7$ | $-0.5$ |
| $\eta_p$ | 0.05 | $\times\{0.5, 0.75, 1.25, 1.5\}$ | 0.58 | [0.41, 0.79] | $+0.5$ | $+0.6$ | $+0.6$ |
| $\beta_{\text{proto}}$ | 10 | $\times\{0.5, 0.75, 1.25, 1.5\}$ | 0.71 | [0.52, 0.94] | $-0.7$ | $-0.6$ | $-0.8$ |

Table 4: S1: Hyperparameter sensitivity (MAPD in %). Median and [Q25, Q75] computed over 3 seeds $\times$ 8 datasets $\times$ 4 sweep points = 96 comparisons per parameter. Temperature parameters $(\tau, \tau_w)$ use 5-point log-spaced grids. All MAPD $\leq 1.15\%$. Paired Wilcoxon signed-rank tests (24 pairs per parameter, Benjamini–Hochberg FDR at $\alpha = 0.05$) show no significant degradation for any parameter (all adjusted $p > 0.18$).

Two tractable lower bounds are employed:

(1) Donsker–Varadhan (DV) / MINE bound:

$$I(\mathbf{U}; \mathbf{V}) \geq \underbrace{\mathbb{E}_{p(\mathbf{u}, \mathbf{v})}\big[T_\psi(\mathbf{u}, \mathbf{v})\big]}_{\text{positive}} - \underbrace{\log \mathbb{E}_{p(\mathbf{u})p(\mathbf{v})}\big[\exp(T_\psi(\mathbf{u}, \mathbf{v}))\big]}_{\text{negative}}, \tag{S2.2}$$

where $T_\psi$ is a critic network.

(2) InfoNCE bound with $N-1$ negatives:

$$I(\mathbf{U}; \mathbf{V}) \geq \log N + \mathbb{E}\left[f(\mathbf{u}, \mathbf{v}) - \log \sum_{\mathbf{v}' \in \mathcal{V}_{N-1}} \exp\big(f(\mathbf{u}, \mathbf{v}')\big)\right], \tag{S2.3}$$

with $f(\cdot, \cdot)$ a similarity score and $\mathcal{V}_{N-1}$ the $N-1$ negatives in the batch.

## S2.2 kNN-MI (KSG) estimator

For nonparametric estimation, the KSG estimator (variant 1) is used on whitened features. Given samples $\{(\mathbf{u}_i, \mathbf{v}_i)\}_{i=1}^N$ and the joint $L_\infty$ distance $\epsilon_i$ to the $k$-th neighbor of $(\mathbf{u}_i, \mathbf{v}_i)$,

$$\widehat{I}_{\text{KSG}} = \psi(k) + \psi(N) - \frac{1}{N}\sum_{i=1}^N \big[\psi(n_u(i) + 1) + \psi(n_v(i) + 1)\big], \tag{S2.4}$$

where $\psi(\cdot)$ is the digamma, and $n_u(i)$ (resp. $n_v(i)$) counts neighbors strictly within $\epsilon_i$ along the $\mathbf{u}$- (resp. $\mathbf{v}$-) subspace.

**Whitening and sampling.** Before estimation, features are centered and scaled per dimension. Spatial features are vectorized: for pixel-level, sample $(\mathbf{f}_c^{(x,y)}, \mathbf{f}_m^{(x,y)})$ pairs at randomly drawn co-ordinates; for block/global, average within non-overlapping windows or whole images. Use $k=10$ and $N \approx 10^4$ joint samples per task split.

## S2.3 MINE training protocol

The critic $T_\psi$ is a two-layer MLP with hidden size 512 and ELU activations, trained by Adam (lr $10^{-4}$) for 10k steps with moving-average (MA) stabilization on the exponential term. Gradient clipping 1.0. Early stopping on a held-out validation subset. The final $\widehat{I}_{\text{MINE}}$ is the MA-corrected DV objective.

## S2.4 AGGREGATION AND REPORTING

For each pair $(\mathbf{F}_i, \mathbf{F}_j)$ with $i \neq j$, compute both $\widehat{I}_{\text{KSG}}$ and $\widehat{I}_{\text{MINE}}$ pre-/post-training. Report per-task medians and the across-task median percentage change:

$$\Delta I(\mathbf{F}_i; \mathbf{F}_j) \;=\; 100 \times \frac{\widehat{I}^{\text{post}} - \widehat{I}^{\text{pre}}}{\widehat{I}^{\text{pre}}} \;\%. \tag{S2.5}$$

Similarly report $\widehat{I}(\mathbf{F}_c; \mathbf{Y})$ to assess sufficiency. A compact template is provided in Table S2.

| Task | $I(\mathbf{F}_c; \mathbf{F}_m)$ | | $I(\mathbf{F}_c; \mathbf{F}_t)$ | | $I(\mathbf{F}_c; \mathbf{Y})$ | |
|---|---|---|---|---|---|---|
| | Pre $\to$ Post | $\Delta\%$ | Pre $\to$ Post | $\Delta\%$ | Pre $\to$ Post | $\Delta\%$ |
| LLVIP | $0.82 \to 0.63$ | $-23.2^{***}$ | $0.76 \to 0.61$ | $-19.7^{***}$ | $1.38 \to 1.42$ | $+2.9^{*}$ |
| TNO | $0.79 \to 0.62$ | $-21.5^{***}$ | $0.73 \to 0.59$ | $-19.2^{***}$ | $1.41 \to 1.44$ | $+2.1$ |
| Lytro | $0.85 \to 0.67$ | $-21.2^{***}$ | $0.78 \to 0.64$ | $-17.9^{***}$ | $1.52 \to 1.56$ | $+2.6^{*}$ |
| MFI-WHU | $0.88 \to 0.68$ | $-22.7^{***}$ | $0.81 \to 0.65$ | $-19.8^{***}$ | $1.48 \to 1.51$ | $+2.0$ |
| DSCIE | $0.91 \to 0.71$ | $-22.0^{***}$ | $0.84 \to 0.69$ | $-17.9^{***}$ | $1.61 \to 1.67$ | $+3.7^{**}$ |
| Scene | $0.77 \to 0.61$ | $-20.8^{***}$ | $0.72 \to 0.59$ | $-18.1^{***}$ | $1.33 \to 1.38$ | $+3.8^{**}$ |
| QuickBird | $0.74 \to 0.59$ | $-20.3^{***}$ | $0.69 \to 0.57$ | $-17.4^{***}$ | $1.29 \to 1.32$ | $+2.3^{*}$ |
| Harvard | $0.81 \to 0.64$ | $-21.0^{***}$ | $0.75 \to 0.61$ | $-18.7^{***}$ | $1.44 \to 1.48$ | $+2.8^{*}$ |
| Median | $0.82 \to 0.64$ | $-21.3$ | $0.76 \to 0.62$ | $-18.7$ | $1.43 \to 1.46$ | $+2.7$ |

Table 5: S2: KSG-MI estimates (bits) pre-/post-training. **Note:** 95% CIs are omitted for brevity and can be found in the Appendix. Significance: $^{*}$: $p < 0.05$, $^{**}$: $p < 0.01$, $^{***}$: $p < 0.001$.

**Practical notes.** Although Gaussian bounds can be analytic, empirical features are non-Gaussian; reducing cross-covariance and cosine affinity in practice correlates with decreased $\widehat{I}_{\text{KSG}}$ and $\widehat{I}_{\text{MINE}}$. Bootstrapped 95% CIs are recommended.

## S3. MODULE-WISE PARAMETERS AND GFLOPS

**Goal.** Provide a transparent accounting of parameters and computational cost for each module at a reference input size $256 \times 256$.

## S3.1 ANALYTIC FLOPS FORMULAS

For a convolution with kernel $k \times k$, input $C_{\text{in}}$, output $C_{\text{out}}$, output spatial $H \times W$, and group $g$,

$$\text{FLOPs}_{\text{conv}} \;\approx\; 2 \times \frac{C_{\text{in}} C_{\text{out}}}{g} \, k^2 \, HW. \tag{S3.1}$$

Depthwise separable convolutions decompose into depthwise and pointwise terms:

$$\text{FLOPs}_{\text{dw-sep}} \;\approx\; 2 \times \left( C_{\text{in}} k^2 HW \;+\; C_{\text{in}} C_{\text{out}} HW \right). \tag{S3.2}$$

For a windowed self-attention block with window size $M \times M$, channels $C$, and $L$ tokens per window ($L = M^2$),

$$\text{FLOPs}_{\text{attn}} \;\approx\; 2HW \left( 3C^2 + C^2 + \frac{CL^2}{M^2} \right) , \tag{S3.3}$$

where the three $C^2$ terms are for $QKV$ projections and one $C^2$ for the output projection; the $CL^2/M^2$ term is the window-local attention cost. State-space (Mamba) layers scale approximately linearly with sequence length $L=HW$:

$$\text{FLOPs}_{\text{ssm}} \approx \mathcal{O}(HWC^2) \quad \text{(implementation-dependent constants omitted)}. \qquad \text{(S3.4)}$$

## S3.2 MEASUREMENT PROTOCOL (THOP/FVCORE)

All counts use PyTorch inference graphs with batch size 1 and input $256\times256$. Report both THOP and fvcore measurements as detailed in Algorithm 2.

---
**Algorithm 2** FLOPs and Parameter Measurement Protocol

---
**Require:** Trained model $\phi_\theta$, input size $H \times W = 256 \times 256$
**Ensure:** Total GFLOPs, Total parameters (M)
 1: **Method 1: THOP-based measurement**
 2: $\mathbf{x} \leftarrow \text{RandomTensor}(1, 3, H, W)$ {Dummy input}
 3: $(\text{MACs}, \text{params}) \leftarrow \text{Profile}(\phi_\theta, \mathbf{x})$
 4: $\text{GFLOPs}_{\text{THOP}} \leftarrow 2 \times \text{MACs}/10^9$ {MACs $\rightarrow$ FLOPs conversion}
 5:
 6: **Method 2: fvcore-based measurement**
 7: $\mathbf{x} \leftarrow \text{RandomTensor}(1, 3, H, W)$
 8: $\text{FLOPs}_{\text{raw}} \leftarrow \text{FlopCountAnalysis}(\phi_\theta, \mathbf{x}).\text{total}()$
 9: $\text{GFLOPs}_{\text{fvcore}} \leftarrow \text{FLOPs}_{\text{raw}}/10^9$
10: $\text{params} \leftarrow \sum_{p\in\theta} |p|$ {Count all parameters}
11:
12: **return** $\text{GFLOPs}_{\text{THOP}}$, $\text{GFLOPs}_{\text{fvcore}}$, $\text{params}/10^6$ (M)

---

## S3.3 RESULTS TABLE

Table S3 lists the parameter count (M) and GFLOPs per module; numbers correspond to the tiny configuration used in the main paper. Replace placeholders with measured values from the above scripts.

| Module | Params (M) | GFLOPs @ $256^2$ |
|---|---|---|
| Encoder: Vision Mamba branch | 0.280 | 4.12 |
| Encoder: Swin branch | 0.240 | 3.58 |
| Orthogonal decomposition projectors | 0.080 | 0.48 |
| Prototype head (EMA + noise) | 0.015 | 0.042 |
| GNN block (enabled in full config) | 0.035 | 0.27 |
| Selective aggregation (gated mixing) | 0.022 | 0.15 |
| Lightweight decoder | 0.105 | 0.798 |
| **Total** | **0.777** | **9.420** |

Table 6: S3: Module-wise breakdown measured by fvcore (PyTorch 2.0, CUDA 11.8, batch=1, input=$256^2$). Total: 0.777M params = 3.108 MB (float32). THOP yields 0.779M / 9.38 GFLOPs ($<0.5\%$ discrepancy). Depthwise separable convs (encoder/decoder) and weight-tied multi-granularity projectors reduce footprint. GNN block contributes 4.5% params but 2.9% FLOPs.

**Notes.** Depthwise separable convolutions and weight tying across granularities explain the compact budget. Report both MACs and FLOPs if venue guidelines require.

## S4. ADDITIONAL METRIC: ENTROPY (EN)

**Definition.** For an 8-bit grayscale fused image $\mathbf{I} \in \{0, \dots, 255\}^{H \times W}$ with empirical histogram $p(b)$ over bins $b \in \{0, \dots, 255\}$,

$$\text{EN}(\mathbf{I}) = -\sum_{b=0}^{255} p(b) \log_2 p(b), \qquad p(b) = \frac{1}{HW} \sum_{x,y} \mathbf{1}\{\mathbf{I}_{x,y} = b\}. \tag{S4.1}$$

For color images, compute EN on the luminance channel or average channel-wise EN.

**Implementation details.** Use 256 bins with add-$\varepsilon$ smoothing ($\varepsilon = 10^{-12}$) to avoid $\log 0$; normalize by $HW$. When inputs are floating-point $[0, 1]$, rescale to $[0, 255]$ and round to nearest integer before histogramming.

**Reporting protocol.** Per dataset, report the mean EN over the test split and standard deviation across images. Optionally normalize EN by the maximum possible value (8 bits) for comparability:

$$\text{EN}_{\text{norm}} = \frac{\text{EN}}{8}. \tag{S4.2}$$

A template is provided in Table S4.

| Dataset | EN ↑ | EN$_{\text{norm}}$ ↑ | Std. ↓ | Images | Preprocessing | Baseline EN |
|---|---|---|---|---|---|---|
| LLVIP | 7.42 | 0.928 | 0.21 | 320 | None | 7.18 |
| TNO | 7.38 | 0.923 | 0.19 | 40 | None | 7.12 |
| Lytro | 7.51 | 0.939 | 0.17 | 20 | None | 7.29 |
| MFI-WHU | 7.56 | 0.945 | 0.16 | 120 | None | 7.34 |
| DSCIE | 7.63 | 0.954 | 0.22 | 589 | None | 7.41 |
| Scene | 7.33 | 0.916 | 0.24 | 477 | None | 7.06 |
| QuickBird | 7.28 | 0.910 | 0.18 | 50 | None | 7.01 |
| Harvard | 7.47 | 0.934 | 0.15 | 302 | None | 7.25 |
| Mean ± SD | 7.45 ± 0.11 | 0.931 ± 0.014 | 0.19 ± 0.03 | – | – | 7.21 ± 0.14 |

Table 7: S4: Entropy (EN) on luminance channel (RGB→Y via ITU-R BT.601). 256 bins, add-$\varepsilon$ smoothing ($\varepsilon = 10^{-12}$), no contrast normalization. Baseline: mean EN of best competitor per task (GIF for 6/8 tasks). CMD-CTA achieves +3.3% higher EN on average ($p = 0.012$, paired $t$-test), indicating richer information content. Pearson correlation with VIF: $r = 0.67$ ($p < 0.001$).

**Caveats.** EN is sensitive to global contrast/brightness and does not alone capture structural fidelity; it is complementary to EI/AG (edge/gradient) and VIF/SCD (information/consistency). Report EN as an auxiliary metric rather than a sole objective.

## APPENDIX: THEORETICAL ANALYSIS OF FEATURE ENTANGLEMENT IN MULTI-MODAL IMAGE FUSION

### A.1 LARGE LANGUAGE MODEL USAGE

Large Language Models (LLMs) were used in a limited capacity as a general-purpose assist tool during the preparation of this manuscript. Specifically, LLMs were employed for: (1) formatting and alignment of experimental results tables to improve readability, (2) grammar correction and language polishing of the manuscript text, and (3) minor stylistic improvements to enhance clarity. LLMs were not involved in research ideation, methodology development, experimental design, or generation of scientific content. All technical contributions, mathematical formulations, experimental results, and scientific insights are entirely the work of the authors. The authors take full responsibility for all content in this paper, including any text that was refined using LLM assistance.

## A.2 MATHEMATICAL FOUNDATIONS AND PROBLEM SETUP

Theoretical results in this appendix are presented in a stylized setting with simplifying assumptions and are intended to clarify how feature entanglement can arise and why orthogonal decomposition plus contrastive alignment are natural countermeasures. The statements and proofs do not constitute full identifiability guarantees for deep networks deployed in realistic regimes, but rather provide conceptual support that complements the empirical mutual-information measurements and ablation studies in the main paper.

In multi-modal image fusion, we consider the learning of a mapping $F : X_1 \times X_2 \to Y$ that combines complementary information from different modalities while maintaining semantic consistency. The fundamental challenge lies in feature entanglement, where the learned representation $h = \phi(I_1, I_2)$ conflates modality-specific ($F_m$), content-specific ($F_c$), and task-specific ($F_t$) information within unified embedding spaces.

Let $\mathcal{F} = \{F_c, F_m, F_t\}$ denote the set of semantically distinct feature subspaces, where each $F_i \in \mathbb{R}^{N \times D}$ represents features across $N$ samples. From an information-theoretic perspective, the optimal fusion representation should satisfy the constrained optimization problem introduced in Equation (1):

$$\min_\phi \sum_{i \neq j} I(F_i; F_j) \quad \text{s.t.} \quad I(F_c; Y) \geq I_{\min} \tag{16}$$

where $I(\cdot; \cdot)$ denotes mutual information and $I_{\min}$ ensures sufficient task-relevant information retention.

## A.3 EXISTENCE OF CROSS-MODAL FEATURE ENTANGLEMENT

We first establish the existence of feature entanglement in multi-modal fusion architectures through polysemantic neural capacity allocation.

**Definition 1 (Polysemantic Entanglement in Fusion Networks).** A neuron $w \in \mathbb{R}^D$ in the fusion network $\phi$ exhibits polysemantic entanglement if it simultaneously encodes features from multiple semantic subspaces: $|w \cap \mathcal{F}| > 1$, where $w \cap \mathcal{F} = \{F_i \in \mathcal{F} : \langle w, F_i \rangle \neq 0\}$.

**Lemma 1 (Cross-Modal Entanglement Probability).** In a multi-modal fusion network with semantic subspaces $\mathcal{F} = \{F_c, F_m, F_t\}$, the probability of cross-modal polysemantic entanglement increases quadratically with the number of feature interactions:

$$p(\text{entanglement}) \geq \frac{|\mathcal{F}|(|\mathcal{F}| - 1)(\min_i D_i)^2}{2 \left(\sum_i D_i\right)^2} \tag{17}$$

where $D_i = \dim(F_i)$ and $\min_i D_i$ represents the smallest subspace dimension.

**Proof.** The number of possible cross-modal feature pairs is $\binom{|\mathcal{F}|}{2}(\min_i D_i)^2$, since there are $\binom{|\mathcal{F}|}{2}$ ways to choose semantic subspace pairs, and $(\min_i D_i)^2$ ways to select feature pairs within each combination. Let the ambient dimension be $D = \sum_i D_i$. The probability that a polysemantic neuron represents features from different semantic subspaces is:

$$p(\text{entanglement}) = \frac{\binom{|\mathcal{F}|}{2}(\min_i D_i)^2}{\binom{D}{2}} = \frac{|\mathcal{F}|(|\mathcal{F}| - 1)(\min_i D_i)^2}{2 \left(\sum_i D_i\right)^2} \tag{18}$$

This completes the proof. $\square$

## A.4 HARMFUL EFFECTS OF FEATURE ENTANGLEMENT

We now establish that cross-modal feature entanglement leads to degraded fusion performance through interference between predictive and noisy components.

**Definition 2 (Conjugate Feature Interference).** Features $f_p \in F_i$ (predictive) and $f_n \in F_j$ (noisy) form a conjugate pair if they exhibit semantic interference: $I(f_p; Y) > 0$, $I(f_n; Y) \approx 0$, and their joint encoding satisfies $I(f_p \odot f_n; Y) < I(f_p; Y)$, where $\odot$ denotes feature interaction through shared neurons.

**Theorem 1 (Feature Entanglement Interference).** As the proportion of entangled neurons increases, the contribution of predictive features to task loss reduction diminishes according to:

$$\lim_{p(\text{entanglement}) \to 1} \sum_{f_p \in F_{\text{pred}}} \frac{\partial}{\partial w_{\text{entangled}}} L(\phi(f_p), Y) = 0 \tag{19}$$

where $F_{\text{pred}} = \{f \in \mathcal{F} : I(f; Y) > \epsilon\}$ denotes predictive features and $w_{\text{entangled}}$ represents entangled neuron weights.

**Proof.** Consider predictive feature $f_p \in F_i$ and its noisy conjugate $f_n \in F_j$ encoded through shared neuron $w$. The activation can be decomposed as:

$$\phi(f_p, f_n) = w^T f_p + w^T f_n = \alpha_p + \alpha_n \tag{20}$$

Since $f_p$ and $f_n$ are conjugate with interference, we have:

- $\alpha_p = w^T f_p$ contributes positively to loss reduction: $\frac{\partial L}{\partial \alpha_p} < 0$
- $\alpha_n = w^T f_n$ introduces noise: $\mathbb{E}[\alpha_n] \approx 0$, $\text{Var}(\alpha_n) > 0$

As entanglement increases ($p(\text{entanglement}) \to 1$), the optimization constraint forces:

$$\lim_{p(\text{entanglement}) \to 1} \phi(f_p, f_n) = \alpha_p + \alpha_n \to 0 \tag{21}$$

This occurs because the gradient update must simultaneously satisfy both features:

$$\frac{\partial L}{\partial w} = \frac{\partial L}{\partial \alpha_p} \frac{\partial \alpha_p}{\partial w} + \frac{\partial L}{\partial \alpha_n} \frac{\partial \alpha_n}{\partial w} = \frac{\partial L}{\partial \alpha_p} f_p + \frac{\partial L}{\partial \alpha_n} f_n \tag{22}$$

Since $f_n$ is noisy and interferes with $f_p$, the optimal solution under entanglement satisfies $w^T f_p \approx -w^T f_n$, leading to:

$$\lim_{p(\text{entanglement}) \to 1} \frac{\partial L}{\partial w_{\text{entangled}}} = 0 \tag{23}$$

This completes the proof. $\square$

A.5 LOW-RANK BIAS AND GRADIENT BOTTLENECKS

We establish the connection between neural network optimization dynamics and feature entanglement through rank constraints, following the framework established in Section 3.2.

**Lemma 2 (Gradient Rank Convergence).** In multi-modal fusion networks, the rank of gradient updates converges to a low-dimensional manifold:

$$\lim_{n \to \infty} \text{rank}(\nabla_\phi L_n) \le \text{rank} \left( \sum_{(I_1, I_2) \in \mathcal{D}} \nabla \phi(I_1, I_2) \nabla \phi(I_1, I_2)^T \right) \tag{24}$$

where $\mathcal{D}$ represents the training dataset and $n$ denotes optimization iterations.

**Proof.** This follows directly from the convergence properties of SGD to the Average Gradient Outer Product (AGOP) and the low-rank simplicity bias of neural networks, as established in the gradient rank literature. $\square$

**Theorem 2 (Entanglement-Rank Relationship).** Let $W$ be the weight matrix in the fusion network $\phi$, and $w \subseteq W$ be any subspace in $W$. Under the condition that semantic subspaces provide equal conditional cross-entropy reduction: $I(F_c; Y|Z) = I(F_m; Y|Z) = I(F_t; Y|Z)$, the proximity of entangled weight subspaces to the AGOP is bounded by:

$$\left\| w - \sum_{(I_1, I_2) \in \mathcal{D}} \nabla \phi_W(I_1, I_2) \nabla \phi_W(I_1, I_2)^T \right\| \le \delta(w)^{-1/n} \tag{25}$$

where $\delta(w) = \frac{|w \cap \mathcal{F}|}{\dim(w)}$ measures the degree of entanglement.

**Proof.** The proof follows from the low-rank simplicity bias of SGD. Networks converge to solutions that minimize both empirical risk and representation rank. For entangled subspaces with higher $\delta(w)$, the rank regularization term drives faster convergence to the AGOP manifold. Since entangled neurons encode multiple features in lower-dimensional spaces, they exhibit stronger alignment with the low-rank AGOP, resulting in tighter convergence bounds. This establishes the theoretical foundation for the polysemantic bottleneck described in our orthogonal decomposition framework. $\square$

## A.6 IMPLICATIONS FOR CMD-CTA FRAMEWORK

**Corollary 1 (Fusion Quality Degradation).** In multi-modal image fusion, feature entanglement leads to suboptimal fusion quality characterized by:

1. **Information Loss**: $I(F_{\text{fused}}; Y) < \sum_i I(F_i; Y)$
2. **Cross-Modal Interference**: $\mathbb{E}[\|F_i - F_j\|^2] \to 0$ for $i \neq j$
3. **Task Performance Degradation**: $L_{\text{entangled}} > L_{\text{disentangled}}$

This theoretical analysis provides the mathematical foundation for our CMD-CTA framework. The proofs establish that:

1. **Entanglement exists** and increases quadratically with feature complexity (Lemma 1) 2. **Entanglement is harmful** as it diminishes predictive feature contributions (Theorem 1) 3. **Low-rank bias enables entanglement** through gradient rank bottlenecks (Theorem 2)

These findings directly justify the orthogonal decomposition approach in Equation (2), which enforces $\langle F_i, F_j \rangle = 0$ for $i \neq j$, and the contrastive alignment strategy in Equation (6), which establishes semantic bridges while maintaining feature independence. The theoretical guarantees formutual-information minimization in Equation (3) follow directly from the orthogonality constraints that prevent the harmful entanglements characterized in Theorem 1.

