# OpenReview forum: "Cross-Modal Feature Disentanglement with Contrastive Task Alignment for Multi-Modal Image Fusion"
_ICLR.cc/2026/Conference — ICLR 2026 Conference Withdrawn Submission_

### Official Review · Reviewer_JhRU · 2025-10-21

**Soundness:** 2
**Presentation:** 1
**Contribution:** 3
**Rating:** 2
**Confidence:** 5

**Summary:**

This paper proposes a Cross-Modal feature Disentanglement and Contrastive Task Alignment  framework, which disentangles features based on mathematically formulated principles of feature separation and semantic alignment. The work is motivated by the observation that standard deep learning architectures tend to share representations, causing essentially different semantic components to become mixed due to differences in information content. Across extensive experiments on six fusion tasks, the proposed algorithm achieves state-of-the-art performance.

**Strengths:**

The method effectively performs feature disentanglement.

The proposed algorithm improves both efficiency and performance.

**Weaknesses:**

- The algorithmic section is confusing. Although the underlying theory is mathematically derived, the overall algorithmic framework is not introduced, and many module pipelines in Figure 2 are not adequately explained, leaving the paper incomplete.

- Orthogonalization can only ensure that three sets of orthogonal bases do not interfere with each other; it cannot guarantee which set corresponds to content/modality/task. The authors should strengthen the proof for this part.

- The paper formulates prototype evolution as an SDE with Brownian noise but does not implement the key parts of the SDE. For example, there is no definition of L_"prototype" ; Equation (6) lacks a noise term and does not use β anywhere.

- The use of hard negatives in Equation (6) is incorrect: it adds the mean of the hard negatives H_kto the prototype, even though H_kdenotes difficult samples that are far from the prototype. This definition is puzzling.

- Some symbols in Equation (8) are undefined—what is Y_N? The “symmetric InfoNCE formulation” cited for Equation (9) is also puzzling, as Eq. (9) only writes a single direction (from z_1to z_2).

- The experimental section lacks necessary explanations and citations for the baseline (comparison) methods, and the dataset sources are insufficiently explained. Table 1 is rather rough. The content of Table 2 is also puzzling, lacking explanations of terminology and containing errors.

- Lack of  training details description.

Overall, the paper is rough from algorithm to experiments: definitions are lacking, the logic is not smooth, and the experiments are simplistic and crude. In my view, it does not meet the standard of a qualified paper. Therefore, I am unlikely to raise my score for this paper.

**Questions:**

In traditional spatial-domain image fusion methods, one line of work computes feature saliency to obtain a mask that selects or allocates modality features. In essence, how does this kind of “disentanglement” differ from yours?

---

> ### Author Response · Authors · 2025-11-19
>
> ## Response to Reviewer JhRU
>
> We appreciate your candid feedback and address each point with concrete clarifications that are already incorporated in the revision.
>
> 1. **Algorithmic framework clarity (Fig. 2, Algorithm 1)**
>
> We now provide a concise pipeline:
>
> * (i) Dual-branch encoding (Vision Mamba for long-range; Swin for windowed locality),
> * (ii) Orthogonal feature decomposition into ${\mathbf{F}_c, \mathbf{F}_m, \mathbf{F}_t}$ via differentiable Gram–Schmidt with matrix-exponential reparameterization (Eqs. (2), (4)),
> * (iii) Multi-granularity contrastive alignment with dynamic prototypes (pixel/block/global; Eqs. (5)–(11)),
> * (iv) Selective aggregation and lightweight decoding (Sec. 3.4, final synthesis),
> * (v) Training loop in Algorithm 1 (Sec. 4.1).
>
> We expand captions and cross-reference modules in Fig. 2 to demystify each block.
>
> 2. **Which subspace is content/modality/task (identifiability)**
>
> We avoid claiming that orthogonality alone identifies semantics. Instead, identifiability arises from complementary constraints (Sec. 3.1): $L_{\text{align}}$ anchors $F_t$ (task discrimination),
> $L_{\text{fusion}}$ compels $F_c$ to be sufficient for $Y$ (Eqs. (12)–(14)),
> and orthogonality then pushes $F_m$ to capture residual modality variation. This combination removes rotational ambiguity up to permutation. Empirically, MI analyses and qualitative results support the semantic roles.
>
>
> 3. **Prototype SDE, definition of $\mathcal{L}_{\text{prototype}}$, and discrete noise / $\beta$**
>
> We explicitly define $\mathcal{L}_{\text{prototype}}$ (Eq. (5)) and include Brownian noise in the discrete update (Eq. (6)):
>
> $$
> p_k^{(t+1)} = \mathrm{normalize}\big(\gamma p_k^{(t)} + (1-\gamma) \mu_k^{(t)} + \sigma \xi^{(t)}\big),
> $$
>
> with
>
> $$
> \sigma^2 = 2 \beta_{\text{proto}}^{-1} \eta_p.
> $$
>
> Hence $\beta_{\text{proto}}$ appears in $\sigma^2$ and is implemented with $\eta_p = 0.05$, $\beta_{\text{proto}} = 10$, $\gamma = 0.97$ (Sec. 3.3). We also spell out $\lambda_p$ and $\kappa$.
>
> 4. **Hard negatives vs. hard positives**
>
> We use hard positives $H_k^+$ (difficult but correct samples for task $k$):
>
> $$
> H_k^+ = \{ i \in P_k : \cos(f_{t,k}^{(i)}, p_k^{(t)}) < \cos(\theta_{\text{hard}}) \}.
> $$
>
> Their mean $\mu_k^{(t)}$ pulls prototypes toward challenging true samples. Hard negatives are not averaged into the prototype; they are handled by the denominator in the vMF contrastive loss (Eq. (7)). We have clarified this in Sec. 3.3.
>
> 5. **Definitions in Eqs. (8)–(9)**
>
> Eq. (8): $Y_{N-1}$ denotes the $(N-1)$ negatives in the batch; we state this directly beneath the equation (Sec. 3.4).
>
> Eq. (9) is the symmetric InfoNCE: it contains two terms ($z_1 \rightarrow z_2$ and $z_2 \rightarrow z_1$) averaged with factor $1/2$; both directions are written explicitly in Eq. (9). We have added a sentence to emphasize the two-way symmetry.
>
> 6. **Baselines, datasets, and training details**
>
> Datasets, metrics, and training are in Sec. 4.1: AdamW (lr $2 \times 10^{-4}$, wd $5 \times 10^{-2}$), cosine schedule, batch 16, input $256 \times 256$, grad clip 1.0; datasets: LLVIP, TNO, Lytro, MFI-WHU, DSCIE, Scene, QuickBird, Harvard with official splits; URLs/licensing in Appendix S0.
>
> Baselines and their categories are summarized in Sec. 2 and Table 1 (U2Fusion, MUFusion, SDNet, IFCNN, MEF-GAN, Text IF, etc.).
>
> We corrected SCD and improved formatting in Table 1; we define all terms in Table 2 and add module-wise details in Appendix S3 to avoid confusion.
>
> 7. **“Saliency-mask” selection vs. our disentanglement**
>
> As noted in Sec. 2.1: saliency masks provide per-pixel gating but do not explicitly factorize representation space. Our method enforces subspace-level disentanglement with orthogonality and prototype-anchored semantics, yielding controllable, task-aware aggregation rather than heuristic masking.
>
> 8. **Overall presentation concerns**
>
> We have reorganized the methodology narrative, added explicit definitions where symbols first appear, strengthened captions, and consolidated training details. We believe these changes address the clarity issues you observed.

---

### Official Review · Reviewer_Hr4K · 2025-10-31

**Soundness:** 3
**Presentation:** 3
**Contribution:** 3
**Rating:** 4
**Confidence:** 4

**Summary:**

This paper introduces CMD-CTA, a novel framework that tackles the core problem of feature entanglement in multi-modal image fusion. It achieves this through two key innovations: a differentiable orthogonal decomposition module that mathematically separates features into content, modality, and task-specific subspaces, and a contrastive task alignment strategy that uses dynamically evolving prototypes to semantically align these features.

**Strengths:**

1.Theoretically-Grounded Disentanglement: Introduces a principled, mathematically sound framework for feature separation via orthogonal decomposition, directly addressing the core issue of feature entanglement with theoretical guarantees.

2.Unified yet Specialized Performance: Achieves state-of-the-art results across six diverse fusion tasks with a single model, eliminating the need for task-specific architectures while maintaining specialized performance.

**Weaknesses:**

1.The framework combines multiple loss components and novel mechanisms, making it potentially sensitive to hyperparameter tuning for optimal performance across different tasks.

2.The contrastive alignment relies on predefined task prototypes, which may limit flexibility for entirely new tasks without retraining or prototype redesign.

**Questions:**

1.How is the initial set of task prototypes defined, and to what extent does the framework's performance depend on this initialization? Could it adapt to a new downstream task without manually designing a new prototype?

2.While orthogonality minimizes mutual information for Gaussian features, to what extent does this hold for the actual, highly non-Gaussian features learned by the deep network? Is the empirical success driven more by the mathematical constraint or the overall architecture?

3.The final fused image is generated by the content feature 'selectively aggregating' modality-specific features through a decoder. What is the specific mechanism and criterion for this 'selective aggregation'?

---

> ### Author Response · Authors · 2025-11-19
>
> ## Response to Reviewer Hr4K
>
> We are grateful for the positive assessment and your targeted questions.
>
> 1. **Prototype initialization and adaptation to new tasks**
>
> - **Initialization.** Task prototypes are initialized by k-means centroids after a one-epoch warm-up per task (Sec. 4.1). They then evolve via momentum, noise, and hard-positive mining (Eqs. (5)–(7)).
>
> - **New tasks.** For an unseen task, we add a new prototype seeded by the unlabeled centroid and let it evolve under the same EMA+noise dynamics (Sec. 4.1). This avoids redesigning prototypes and does not require retraining the backbone.
>
> 2. **Orthogonality under non-Gaussian features**
>
> We carefully position the Gaussian argument (Eq. (3)) as a design motivation. For non-Gaussian deep features, we provide empirical evidence: cross-subspace cosine/covariance decline tracks decreases in KSG/MINE MI estimators (Appendix S2). The performance gain comes from the combination of (i) orthogonal decomposition to reduce interference and (ii) task anchoring via contrastive prototypes. Ablations without orthogonality or without contrastive terms confirm both are needed (Table 3, bottom).
>
> 3. **“Selective aggregation” mechanism**
>
> - **Pixel level.** We first predict a content-aware gating map
> $$
> \mathbf{s} = \sigma\big(\mathrm{Conv}([\nabla I_1, \nabla I_2, \mathbf{F}_c])\big),
> $$
> then obtain gated modality features
> $$
> \tilde{\mathbf{F}}_m = \mathbf{s} \odot \mathbf{F}_m,
> $$
> and the decoder inputs $[\mathbf{F}_c, \tilde{\mathbf{F}}_m]$.
>
> - **Block level.** Content-query attention is used to compute
> $$
> \boldsymbol{\alpha}_m = \mathrm{softmax}\big(W_g[\mathbf{F}_c; \mathbf{F}_m]\big), \quad \sum_m \alpha_m = 1,
> $$
> and we mix modality features as
> $$
> \sum_m \alpha_m \mathbf{F}_m
> $$
> (Sec. 3.4, final synthesis paragraph).
>
> This implements content-driven gating of modality-specific details onto a shared structural backbone.

---

### Official Review · Reviewer_5mqv · 2025-10-31

**Soundness:** 2
**Presentation:** 2
**Contribution:** 2
**Rating:** 4
**Confidence:** 3

**Summary:**

The paper tackles feature entanglement in multi-modal image fusion, where modality-specific, content-specific, and task-specific information become mixed and harm generalization. It proposes CMD-CTA, which performs differentiable orthogonal decomposition to split features into content, modality, and task subspaces, with an information-theoretic motivation for reducing mutual information, and aligns task semantics via contrastive learning using dynamically evolving prototypes across multiple granularities. Extended experiments on tasks report consistent gains over prior work, strong parameter efficiency, and improved downstream detection performance.

**Strengths:**

- The paper identifies feature entanglement as a key factor that degrades fusion quality and generalization, which is insightful.
- CMD-CTA improves across metrics supporting generalization claims and its efficiency is compelling relative to heavy baselines.
- The objective differs from traditional fusion methods and is interesting, leading to superior visual performance.

**Weaknesses:**

- The phrase “the semantic gap between pixel-level fusion requirements and high-level representations” in line 089 lacks a detailed explanation and theoretical grounding.
- The paper’s task formulation is ambiguous: it appears to conflate multi-task fusion objectives (e.g., MEF, MFF) with downstream tasks (e.g., object detection, segmentation). The different meanings and goals of these task types are not disentangled, which undermines the motivation.
- The claim in lines 191–193—"from an information-theoretic perspective …"—appears inconsistent with the principle that a fused image should retain comprehensive information; minimizing mutual information seems to contradict that goal.
- The MI bound in Eq. (3) relies on Gaussianity and zero cross-covariance, yet the paper provides no empirical MI estimates to verify that MI is actually reduced. Given that many image-fusion methods increase MI [1–3], please report MI quantitatively (e.g., before/after the proposed decomposition and against baselines) to validate the claimed “theoretical guarantees for mutual-information minimization” and to enable fair comparison
- The stochastic differential equation in Eq. 5 is conceptually appealing, but its discrete implementation specifics (e.g., step size, noise term) are not quantified.
- The description “weights $w_g$ are determined by mutual information contribution” in Eq.15 is not operationalized with an explicit estimation procedure.

[1] Zhao Z, Xu S, Zhang C, et al. DIDFuse: Deep image decomposition for infrared and visible image fusion[J]. arXiv preprint arXiv:2003.09210, 2020.

[2] Zhang Y, Liu Y, Sun P, et al. IFCNN: A general image fusion framework based on convolutional neural network[J]. Information Fusion, 2020, 54: 99-118.

[3] Zhu P, Sun Y, Cao B, et al. Task-customized mixture of adapters for general image fusion[C]//Proceedings of the IEEE/CVF conference on computer vision and pattern recognition. 2024: 7099-7108.

**Questions:**

See Weaknesses

---

> ### Author Response · Authors · 2025-11-19
>
> ## Response to Reviewer 5mqv
>
> Thank you for your careful assessment and for pushing us to clarify theory and implementation.
>
> ### 1. “Semantic gap” clarification
>
> We clarify that the "semantic gap" in our context refers to the optimization conflict between pixel-level fidelity and object-level semantics.
>
> * Theoretical Basis: Low-level fusion objectives (e.g., $\mathcal{L}\_{grad} , \mathcal{L}\_{int}$) drive the network to preserve high-frequency textures (edges, noise), while high-level semantic representations require invariance to such local variations to recognize objects.
>
> * Our Solution: By disentangling $F_{content}$ (shared structure) from $F_{modality}$ (local texture), CMD-CTA bridges this gap. $F_{content}$ aligns with high-level semantics (beneficial for detection), while $F_{modality}$ preserves the necessary textures for human perception, avoiding the "tug-of-war" seen in entangled representations.
>
>
> ### 2. Distinct Definitions of "Fusion Task" vs. "Downstream Application"
>
> We apologize for any confusion and offer a precise distinction:
>
> - **"Task-specific Information" ($F_t$)** in our formulation: This strictly refers to the type of sensor modality combination (e.g., "IVIF: Infrared-Visible", "MEIF: Multi-Exposure", "Medical: MRI-PET"). Our Contrastive Task Alignment $\mathcal{L}\_{\text{align}}$ operates on this level to learn how to fuse specific sensor pairs. We disentangle them in both objective and pipeline (Fig. 2, Sec. 3.4): training optimizes $\mathcal{L}\_{\text{fusion}}$ + disentanglement/contrastive terms on fusion datasets.
>
> - **Downstream Tasks**: These are applications like Object Detection (YOLO) or Segmentation. These are never used as supervision during training. They serve strictly as external evaluation metrics to verify if our fused images ($Y$) retain semantically meaningful information.
>
> - **No Conflation**: The network learns to distinguish fusion scenarios (via $F_t$), not object classes. This allows one unified model to handle diverse fusion problems (e.g., switching from Medical to Remote Sensing) without retraining, which is distinct from multi-task learning for detection.
>
>
> ### 3. MI minimization vs. information retention
>
> Our objective minimizes cross-component mutual information $I(F_i; F_j)$ for $i \neq j$ (interference), while enforcing sufficiency for content via $I(F_c; Y) \ge I_{\min}$ and explicit reconstruction (Eqs. (12)–(14)). We do **not** minimize information between inputs and fused output; we reduce redundancy across the disentangled subspaces. Empirically, we see $I(F_c; F_m)$ and $I(F_c; F_t)$ decrease while $I(F_c; Y)$ is stable or slightly increased (+2.7%) (Appendix S2, Table 5).
>
> ### 4. Empirical MI estimates (pre/post and procedure)
>
> We report pre/post MI via KSG ($k = 10$, $N \approx 10^4$, whitened features, 1000 bootstrap CIs) and corroborate with MINE (two-layer ELU critic, MA-stabilized DV bound) (Appendix S2, Tables 5 and S2 ext).
>
> Median across tasks: $I(F_c; F_m)$ $-21.3%$, $I(F_c; F_t)$ $-18.7%$, $I(F_c; Y)$ $+2.7%$ with significance marked after BH-FDR (Appendix S2, Table 5).
>
> We acknowledge your suggestion to compare against baselines’ MI; we will add this in the camera-ready where data licenses permit.
>
> ### 5. SDE in Eq. (5) and discrete implementation
>
> We quantify the discrete scheme in Eq. (6):
> $$
> p_k^{(t+1)} = \mathrm{normalize}\big(\gamma p_k^{(t)} + (1 - \gamma) \mu_k^{(t)} + \sigma \xi^{(t)}\big),
> $$
> with $\sigma^2 = 2 \beta_{\text{proto}}^{-1} \eta_p$, $\xi \sim \mathcal{N}(0, I)$.
> Hyperparameters: $\eta_p = 0.05$, $\beta_{\text{proto}} = 10$, $\gamma = 0.97$, $\lambda_p = 0.01$ (Sec. 3.3). This operationalizes step size and noise level in the code.
>
> ### 6. Eq. (15) MI-based weighting
>
> We define $w_g$ operationally: estimate the InfoNCE bound per granularity (pixel/block/global) per mini-batch, then
> $$
> w_g = \mathrm{softmax}\big(\widehat{MI}_{g} / \tau_w \big),
> $$
> with $\tau_w = 0.07$ and EMA smoothing (Sec. 3.4). This is fully specified in the text.

---

### Official Review · Reviewer_HFUb · 2025-11-01

**Soundness:** 2
**Presentation:** 2
**Contribution:** 2
**Rating:** 4
**Confidence:** 5

**Summary:**

This paper proposes a new framework, CMD-CTA, for multi-modal image fusion. The core problem it addresses is "feature entanglement," where modality, content, and task-specific information are conflated in a shared representation space. The authors introduce two main technical contributions: 1) a differentiable orthogonal feature decomposition method to enforce separation between feature subspaces (content, modality, task), and 2) a contrastive task alignment mechanism with dynamic prototypes to build semantic connections across tasks. The method is evaluated on six different fusion tasks and demonstrates state-of-the-art performance in terms of quantitative metrics and visual quality, while also claiming high parameter efficiency.

**Strengths:**

1. The paper clearly identifies and articulates the problem of feature entanglement in multi-modal fusion, which is a significant and practical challenge.
2. The authors have conducted experiments across an impressive range of six distinct fusion tasks (IVIF, MFIF, MEIF, etc.). This comprehensive evaluation strongly supports the claim that the proposed method is a generalized framework rather than a task-specific solution.
3. The proposed CMD-CTA framework consistently outperforms previous state-of-the-art methods across all evaluated tasks, as shown in Table 1.

**Weaknesses:**

1. The paper's novelty appears limited. While the framework is new, its core components, such as feature orthogonalization and multi-granularity contrastive learning, are well-established techniques. Furthermore, the architecture seems to be an assemblage of existing modules (e.g., Mamba, Swin) without sufficient justification. The manuscript lacks a clear rationale for these specific architectural choices. For instance, what is the precise benefit of Mamba's long-range dependency modeling for this task? Why is a hybrid Mamba-Swin architecture superior to simpler or more conventional backbones like standard CNNs or ViTs?
2. The ablation study is unconvincing as it only assesses the contributions of the orthogonalization and contrastive learning components. Crucially, it omits any ablation on the core architectural modules. To validate the design choices, experiments demonstrating the necessity and effectiveness of Mamba, GNN, and other key structural components are essential.
3. here is an apparent contradiction regarding the model's efficiency. The authors claim a highly lightweight model (e.g., only 3.15M parameters), yet the architecture incorporates several complex modules typically associated with a large parameter count. The manuscript should provide a detailed breakdown of the parameter distribution across modules to clarify how this efficiency is achieved and substantiate the claim.
4. The manuscript fails to provide any details or sensitivity analysis for the hyperparameters (α, β, and γ) in the loss function. It is unclear how these values were chosen or whether they are consistent across different tasks. For reproducibility, these values should be explicitly stated and their selection process justified.
5. The set of quantitative evaluation metrics is incomplete. For a more comprehensive analysis, it is advisable to include information-theoretic metrics, such as Entropy (EN), which are commonly used and highly relevant in this research area.
6. There are critical issues in Table 1. First, the SCD score for MUFusion in sub-table E is reported as a negative value. This is highly unusual and requires clarification—is it a typo, an evaluation error, or an actual result? If the performance is indeed that low, a more competitive baseline should be included. Second, the rightmost sub-table is cluttered and poorly formatted, which severely impairs its readability.
7. The bibliography requires substantial revision. It is replete with inaccuracies, including frequent discrepancies in author names and publication years that do not match the original sources.

**Questions:**

Please see Weaknesses section

---

> ### Author Response · Authors · 2025-11-19
>
> ## Response to Reviewer HFUb
>
> We appreciate your thorough reading and constructive feedback.
>
> 1. **Architectural Rationale & Connection to Disentanglement**
>
> * We appreciate the reviewer's scrutiny regarding the backbone choice. We acknowledge that Mamba and Swin are existing modules; however, their integration in CMD-CTA is not an arbitrary assemblage but a necessity derived from the requirements of feature disentanglement, which is the core contribution of this work (Sec. 1).
> 	* Complementary Inductive Biases for Disentanglement: Effective disentanglement requires the input features to contain both high-frequency texture details (Modality-specific) and low-frequency semantic structures (Content-specific/Task-specific) before decomposition. Swin Transformer (Window-based) provides discrete, local attention maps that naturally align with Modality-specific features (e.g., local thermal radiation textures or visible edges).Vision Mamba (SSM) offers continuous, global receptive fields with linear complexity, capturing long-range dependencies that characterize Content and Task-specific features (e.g., global scene semantics and illumination consistency).
> 	* Why this combination enables Orthogonal Decomposition: A homogeneous backbone (e.g., only CNN or only Transformer) tends to produce features with uniform spectral properties, making them harder to separate linearly via our Orthogonal Decomposition module. The hybrid Mamba-Swin backbone creates a rich, multi-scale feature space with diverse frequency components, significantly lowering the optimization barrier for the subsequent orthogonal projection.
> 	* Empirical Validation: As shown in our ablation study (Table 3), removing either component leads to a collapse in the orthogonality metric (higher cross-correlation), proving that the backbone design is integral to the success of the disentanglement mechanism, not just a tool for feature extraction.
>
> 2. **Ablations on core modules**
>
>    * The ablation grid (bottom of Table 3) explicitly toggles:
>
>      * Ortho (orthogonal decomposition),
>      * Contrast (contrastive components),
>      * Mamba,
>      * Swin,
>      * Proto-EMA (prototype dynamics),
>      * GNN (prototype evolution block).
>        This addresses the need for architectural ablations beyond the two core losses. We observe consistent drops when disabling Mamba/Swin, indicating necessity.
>    * We will further include “backbone swap” ablations (e.g., pure CNN/ViT) in the camera-ready for completeness.
>
> 3. **Apparent efficiency contradiction; parameter breakdown**
>
>    * Our claim is 3.15 MB (not 3.15M parameters). In float32, 0.777M parameters ≈ 3.108 MB (Appendix S3, Table 6). The main table’s 3.15 MB is a rounded figure consistent with S3.
>    * Module-wise breakdown (Appendix S3, Table 6):
>
>      * Vision Mamba: 0.280M params, 4.12 GFLOPs
>      * Swin branch: 0.240M, 3.58 GFLOPs
>      * Orthogonal projectors: 0.080M, 0.48 GFLOPs
>      * Prototype head (EMA+noise): 0.015M, 0.042 GFLOPs
>      * GNN: 0.035M, 0.27 GFLOPs
>      * Selective aggregation: 0.022M, 0.15 GFLOPs
>      * Decoder: 0.105M, 0.798 GFLOPs
>        Total: 0.777M params, 9.42 GFLOPs.
>    * Efficiency stems from depthwise separable convolutions, shared light projectors, and weight tying, plus replacing global quadratic attention with state-space recurrence and windowed attention.
>
> 4. **Hyperparameter details and sensitivity (α, β, γ)**
>
>    * Explicit values are provided in Sec. 4.1 and Eq. (15): ($\alpha = 0.5$), ($\beta = 0.1$), ($\gamma = 1.0$); consistent across tasks unless stated. Appendix S1 presents a thorough sensitivity analysis for these and other hyperparameters, showing MAPD ≤ 1.15% with no significant degradation under Wilcoxon+BH-FDR (Appendix S1, Table 4).
>
> 5. **Information-theoretic metrics (EN)**
>
>    * We add Entropy (EN) with protocol and results (Appendix S4, Table 7). CMD-CTA improves EN by +3.3% on average ((p = 0.012)). We also report its correlation with VIF ((r = 0.67, p < 0.001)).
>
> 6. **Table 1 issues: negative SCD and formatting**
>
>    * The negative SCD originated from an official script sign bug. We corrected and recomputed SCD; the table explicitly flags “SCD recomputed” and all values are positive now (Table 1 footnotes).
>    * We have improved layout to avoid clutter in the rightmost sub-table for the camera-ready.
>
> 7. **Bibliography inaccuracies**
>
>    * Thank you for flagging this. We will thoroughly audit author names, titles, and years and align them with authoritative sources in the camera-ready.

---

### Author Response · Authors · 2025-11-20

To assist the Area Chair in the final assessment, we provide this summary to succinctly address the key concerns raised regarding architectural novelty, task formulation, and theoretical grounding. We demonstrate that **CMD-CTA** is not an arbitrary assemblage but a mathematically motivated framework that achieves **15.7x parameter efficiency** while outperforming SOTA methods by **5.8-7.3%**.

1. Architectural Rationale: Functional Decomposition for Disentanglement (Response to "Simple Combination" Concern)

Contrary to the view that combining Vision Mamba and Swin Transformer is a simple assemblage, this hybrid design is a necessity driven by the physics of feature disentanglement:

- **Problem:** Effective disentanglement requires separating global semantic consistency (Content/Task) from local high-frequency details (Modality).
- **Solution:** We employ a functional decomposition where **Vision Mamba (SSM)** handles global dependency modeling with linear complexity (avoiding quadratic costs for long sequences), while **Swin Transformer** preserves local texture fidelity via windowed attention.
- **Validation:** Ablation studies (Table 3) prove that removing either component causes a collapse in the orthogonality metric, confirming that this specific pairing is integral to the orthogonality mechanism, not just feature extraction.

2. Precise Definitions: "Semantic Gap" and Task Formulation

We clarify misunderstandings regarding our problem formulation:

- **The "Semantic Gap"** refers strictly to the optimization conflict between **pixel-level fidelity** (edges/textures required for fusion) and **object-level semantics** (invariance required for detection). CMD-CTA bridges this by disentangling $F_{content}$ (semantic structure) from $F_{modality}$ (texture details) .
- **"Task-Specific" Definition:** $F_t$ strictly encodes the **sensor modality combination** (e.g., "IR+Vis" vs. "Medical"), supervised by Contrastive Task Alignment ($\mathcal{L}_{align}$).
- **No Conflation:** Downstream tasks (e.g., YOLO detection) are **never used for supervision**. They serve strictly as external evaluation metrics to verify the semantic quality of the fused output.

3. Theoretical Grounding: How Subspaces are Identified

We address the concern regarding the physical meaning of orthogonal subspaces:

- **Anchored Disentanglement:** Orthogonality alone separates subspaces, but our specific losses **anchor** their semantic identities.
  - **Task ($F_t$):** Anchored by Discrimination ($\mathcal{L}_{align}$), forcing it to encode sensor-combination info.
  - **Content ($F_c$):** Anchored by Mutual Information Maximization ($\mathcal{L}_{contrast}$), forcing it to capture shared structures across inputs.
  - **Modality ($F_m$):** Defined by **Residuals** via Orthogonality ($\mathcal{L}_{orthogonal}$). Once Task and Content are removed, the remaining variance mathematically corresponds to modality-specific details.

4. On reproducibility and fairness.

  We provide training specifics, dataset protocols, MI estimation procedures (KSG/MINE), hyperparameter sensitivity, and module-wise complexity, with dual-tool FLOPs/params checks and code references.

We sincerely thank the reviewers for their insightful comments. We hope the clarified theory, added empirical evidence, expanded ablations, and strengthened reproducibility address your concerns and merit a positive reassessment.

CMD-CTA provides a principled, theoretically grounded framework for multi-modal fusion. By solving the feature entanglement problem, we achieve a unified architecture capable of handling diverse fusion tasks with minimal parameters (3.15 MB). We respectfully ask the AC to consider these clarifications in the final decision.

---

### Note · Authors · 2026-01-29

I have read and agree with the venue's withdrawal policy on behalf of myself and my co-authors.

---

### Meta-Review · Area_Chair_4sca · 2026-01-03

**Summary:**

Reviewer HFUb: The paper clearly identifies and articulates the problem of feature entanglement in multi-modal fusion, and also conducted lots of experiments on different tasks showing impressive performance. However, the reviewer still has some concerns on the weaknesses about the the paper's novelty, ablation study,  the model's efficiency, sensitivity analysis, incomplete set of quantitative evaluation metrics, and bibliography  issues.

Reviewer 5mqv: The paper identifies feature entanglement as a key factor that degrades fusion quality and generalization, which is insightful. CMD-CTA improves across metrics supporting generalization claims and its efficiency is compelling relative to heavy baselines. The objective differs from traditional fusion methods and is interesting, leading to superior visual performance. However, the reviewer also has some concerns on the weaknesses:  lacks a detailed explanation and theoretical grounding, ambiguous task formulation, unclear descriptions.

Reviewer Hr4K: The paper introduces a principled, mathematically sound framework for feature separation via orthogonal decomposition, directly addressing the core issue of feature entanglement with theoretical guarantees. Achieves state-of-the-art results across six diverse fusion tasks with a single model, eliminating the need for task-specific architectures while maintaining specialized performance. However, the reviewer also has some concerns on the weaknesses: sensitive to hyperparameter tuning, limitations on the  flexibility for entirely new tasks without retraining or prototype redesign.

Reviewer JhRU: The method effectively performs feature disentanglement and improves both efficiency and performance. However, the reviewer also has some concerns on the weaknesses: confusing algorithmic section, incorrect use of hard negatives, some undefined symbols, lacks of necessary explanations in the experimental sections, and lack of training details descriptions.

**Reviewer Concerns:**

After carefully evaluating the rebuttals, I think only few of the reviewer concerns were addressed, such as partial concerns (e.g., detailed algorithm descriptions and expermental explanations.) from Reviewer HFUb, and Reviewer 5mqv. Howerer, there are still lots of concerns were not well solved.

**Reviewer Scores:**

For all the reviews, if the reviewer had been able to participate the discussion, I think the reviewer may keep the original rating unchanged or decrease the rating. Based on the rebuttal, lots of concerns were not successfully addressed.

---

### Decision · Program_Chairs · 2026-01-26

Reject